# Revisiting Discrete Soft Actor-Critic

**Haibin Zhou**                                             *haibinzhou@tencent.com*
*Tencent Inc.*

**Tong Wei**                                             *wt22@mails.tsinghua.edu.cn*
*Tsinghua University*

**Zichuan Lin**                                             *zichuanlin@tencent.com*
*Tencent Inc.*

**Junyou Li**                                             *junyouli@tencent.com*
*Tencent Inc.*

**Junliang Xing**                                             *jlxing@tsinghua.edu.cn*
*Tsinghua University*

**Yuanchun Shi**                                             *shiyc@tsinghua.edu.cn*
*Tsinghua University*

**Li Shen**                                             *mathshenli@gmail.com*
*Sun Yat-sen University*

**Chao Yu**                                             *yuchao3@mail.sysu.edu.cn*
*Sun Yat-sen University*

**Deheng Ye**[*]                                             *dericye@tencent.com*
*Tencent*

**Reviewed on OpenReview:** *https://openreview.net/forum?id=EUF2R6VBeU*

## Abstract

We study the adaptation of Soft Actor-Critic (SAC), which is considered as a state-of-the-art reinforcement learning algorithm, from continuous action space to discrete action space. We revisit vanilla discrete SAC, i.e., SAC for discrete action space, and provide an in-depth understanding of its Q value underestimation and performance instability issues when applied to discrete settings. We thereby propose Stable Discrete SAC (SD-SAC), an algorithm that leverages entropy-penalty and double average Q-learning with Q-clip to address these issues. Extensive experiments on typical benchmarks with discrete action space, including Atari games and a large-scale MOBA game, show the efficacy of SD-SAC. Our code is at: `https://github.com/coldsummerday/SD-SAC.git`.

## 1 Introduction

In the conventional model-free reinforcement learning (RL) paradigm, an agent can be trained by learning an approximator of action-value (Q) function (Mnih et al., 2015; Bellemare et al., 2017). The class of actor-critic algorithms (Mnih et al., 2016; Fujimoto et al., 2018) evaluates the policy function by approximating the value function. Motivated by maximum-entropy RL (Ziebart et al., 2008; Rawlik et al., 2012; Abdolmaleki et al., 2018), soft actor-critic (SAC) (Haarnoja et al., 2018a) introduces action entropy in the framework of

---

[*]Corresponding author.

actor-critic to balance exploitation and exploration. It has achieved remarkable performance in a range of environments with continuous action spaces (Haarnoja et al., 2018b), and is considered as the state-of-the-art algorithm for domains with continuous action space, e.g., Mujoco (Todorov et al., 2012).

However, while SAC solves problems with continuous action space, it cannot be directly applied to discrete domains since it relies on the reparameterization of Gaussian policies to sample actions, in which the action in discrete domains is categorical. Soft-DQN (Vieillard et al., 2020) provides a simple way to discretize SAC by adopting the maximum-entropy RL to DQN (Mnih et al., 2013). However, Soft-DQN utilizes only a Q-value parametrization to bypass the policy parameterization. Another discretization of the continuous action output and Q value in vanilla SAC is suggested by previous work (Christodoulou, 2019) to adapt SAC to discrete domains, resulting in discrete SAC (DSAC). However, it is counter-intuitive that the empirical experiments in subsequent efforts (Xu et al., 2021) indicate that discrete SAC performs poorly in discrete domains, e.g., Atari games. We believe that the idea of maximum entropy RL applies to both discrete and continuous domains. However, extending the maximum-entropy-based SAC algorithm to discrete domains still lacks a commonly accepted practice in the community. Therefore, in this paper, similar to the motivation of DDPG (deep deterministic policy gradient) (Lillicrap et al., 2016), which adapts DQN (deep Q networks) (Mnih et al., 2013) from discrete action space to continuous action space, we aim to optimize SAC algorithm for discrete domains.

Previous studies (Xu et al., 2021; Wang & Ni, 2020) have analyzed the reasons for the performance disparity of SAC between continuous and discrete domains. Reviewing from the perspective of automating entropy adjustment, an unreasonable setting of target-entropy for temperature $\alpha$ may break the SAC value–entropy trade-off (Wang & Ni, 2020; Xu et al., 2021). Furthermore, the function approximation errors of Q-value are known to lead to estimation bias and hurt performance in actor-critic methods (Fujimoto et al., 2018). To avoid overestimation bias, both discrete SAC and continuous SAC resort to clipped double Q-learning (Fujimoto et al., 2018) for actor-critic algorithms. On the contrary, using the lower bound approximation to the critic can lead to underestimation bias, which makes the policy fall into pessimistic underexplored, as pointed by (Ciosek et al., 2019; Pan et al., 2020), mainly when the reward is sparse. However, existing works only focus on continuous domains (Ciosek et al., 2019; Pan et al., 2020), while SAC for discrete cases remains less explored.

In addition to the abovementioned issues, we conjecture that discrete SAC fails also due to the absence of policy update constraints. Intuitively, the unstable training causes a shift in the Q function distribution and policy entropy, which generates a rapidly changing target for the critic network due to the soft Q-learning objective. Meanwhile, the critic network in SAC needs time to adapt to the oscillating target process, exacerbating policy instability.

To address the above challenges, we first design test cases to replicate the failure modes of vanilla discrete SAC, exposing its inherent weaknesses regarding training instability and Q-value underestimation. Then, accordingly, we propose Stable Discrete SAC (SD-SAC) to stabilize the training. We develop an entropy penalty on the policy optimization objective to constrain policy updates. We also develop double average Q-learning with Q-clip to confine the Q value within a reasonable range. We use Atari games (the default testbed for the RL algorithm for discrete action space) to verify the effectiveness of our optimizations. We also deploy our method to the Honor of Kings 1v1 game, a large-scale MOBA game used extensively in recent RL advances (Ye et al., 2020b;c;a; Wei et al., 2022), to demonstrate the scale-up capacity of our Stable Discrete SAC.

To sum up, our contributions are:

- We pinpoint two failure modes of discrete SAC, regarding training instability and underestimated Q values. We find that the underlying causes are the environment's deceptive rewards and SAC's double Q learning respectively.

- To alleviate the training instability issue, we propose entropy-penalty to constrain the policy update in discrete SAC.

- To deal with the underestimation bias of Q value in discrete SAC, we propose double average Q-learning with Q-clip to estimate the state-action value.

With the above contributions, we have obtained the Stable Discrete SAC (SD-SAC) algorithm. Extensive experiments on Atari games and a large-scale MOBA game show SD-SAC's superiority compared to baselines, with a 68% improvement of normalized scores in Atari and around 100% ELO increase in the Honor of Kings 1v1 game environment.

## 2 Related Work

**Adaptation of Action Space**. The most relevant works to this paper are vanilla discrete SAC (Christodoulou, 2019), TES-SAC (Xu et al., 2021) and Soft-DQN (Vieillard et al., 2020). Discrete SAC replaces the Gaussian policy with a categorical one and discretizes the Q-value output to adapt SAC from continuous to discrete action space. However, as we will point out, a direct discretization of SAC will have specific failure modes with poor performance. TES-SAC proposes a new scheduling method for the target entropy parameters in discrete SAC. Soft-DQN has discretized SAC by adopting the maximum-entropy RL to DQN, utilizing only a Q value parametrization and directly applies a softmax operation to the Q-values to take action.

**Q Estimation**. Previous works (Fujimoto et al., 2018; Ciosek et al., 2019; Pan et al., 2020; Duan et al., 2021) have already expressed concerns about the estimation bias of Q value for SAC. SD3 (Pan et al., 2020) proposes to reduce the Kurtosis distribution of Q approximately by using the softmax operator on the original Q value output to reduce the overestimation bias. OAC (Ciosek et al., 2019) constrains the Q value approximation objective by calculating the upper and lower boundaries of two Q-networks. Distributional SAC (Duan et al., 2021) replaces the Q learning target with the expected reward sum obtained from the current state to the end of the episode and uses a multi-frame estimates target to reduce overestimation. Maxmin Q-learning (Lan et al., 2020) controls estimation bias by minimizing the complete ensemble in the target. MME (Han & Sung, 2021) extends max-min operation to the entropy framework to adapt to SAC. REM (Agarwal et al., 2020) ensembles Q-value estimations with a random convex combination to enhance generalization in the offline setting. REDQ (Chen et al., 2021b) reduces the estimation bias by minimizing a random subset of Q-functions. AEQ (Gong et al., 2023) adjusts the estimation bias by using the mean of Q-functions minus their standard deviation. However, little research is on discrete settings. Our approach focuses on reducing the underestimation bias for the double Q-estimators to enhance exploration.

**Performance Stability**. Flow-SAC (Ward et al., 2019) applies a technique called normalizing flows policy on continuous SAC leading to the finer transformation that improves training stability when exploring complex states. However, applying normalizing flows to discrete domains will cause a degeneracy problem (Horvat & Pfister, 2021), making it difficult to transfer to discrete actions. SAC-AWMP (Hou et al., 2020) improves the stability of the final policy by using a weighted mixture to combine multiple policies. Based on this method, the cost of network parameters and inference speed is significantly increased. ISAC (Banerjee et al., 2022) increases SAC stability by mixing prioritized and on-policy samples, enabling the actor to repeat learns states with drastic changes. Repeatedly learning priority samples, however, runs the risk of settling into a local optimum. By comparison, our method improves policy stability in case of drastic state changes with an entropy constraint.

## 3 Preliminaries

This section briefly overviews the symbol definitions of SAC for discrete action space. Following the maximum entropy framework, SAC adds an entropy term $\mathcal{H}(\pi(\cdot \mid s))$ as regularization to the policy gradient objective:

$$\pi^* = \operatorname*{argmax}_{\pi} \sum_{t=0}^{T} \left[ \gamma^t \mathop{\mathbb{E}}_{\substack{s_t \sim p \\ a_t \sim \pi}} [r(s_t, a_t) + \alpha \mathcal{H}(\pi(\cdot \mid s_t))] \right], \tag{1}$$

$$\mathcal{H}(\pi(\cdot \mid s)) = -\sum_a \pi(a \mid s) \log \pi(a \mid s) = \mathop{\mathbb{E}}_{a \sim \pi(\cdot \mid s)} [-\log \pi(a \mid s)] \tag{2}$$

where $\pi$ is a policy, $\pi^*$ is the optimal policy, and $\alpha$ is the temperature parameter that determines the relative importance of the entropy term versus the reward $r$, thus controls the stochasticity of the optimal policy.

**Soft Bellman Backup** The soft Q-function, parametrized by $\theta$, is updated by reducing the soft Bellman error as described in the next subsection:

$$J_Q(\theta) = \frac{1}{2}\left(r(s_t, a_t) + \gamma V(s_{t+1}) - Q_\theta(s_t, a_t)\right)^2, \tag{3}$$

where $V(s_t)$ defines the soft state value function, which represents the expected reward estimate that policy obtains from the current state to the end of the trajectory.

$$V(s_t) = \mathbb{E}_{a_t \sim \pi}[Q_\theta(s_t, a_t) - \alpha \log(\pi(a_t \mid s_t))]. \tag{4}$$

Soft actor-critic minimizes soft Q-function with final soft Bellman error:

$$J_Q(\theta) = \mathbb{E}_{(s_t, a_t) \sim D}[\frac{1}{2}(Q_\theta(s_t, a_t) - (r(s_t, a_t) + \gamma \mathbb{E}_{s_{t+1} \sim p(\cdot|s_t, a_t)}[V(s_{t+1})]))^2], \tag{5}$$

where $D$ is a replay buffer replenished by rollouts of the policy $\pi$ interacting with the environment. In the implementation, SAC (Haarnoja et al., 2018a) uses the minimum of two delayed-update target-critic network outputs as the soft bellman learning objective to reduce overestimation. The formula is expressed as

$$V(s_{t+1}) = \min_{i=1,2} \mathbb{E}_{a_t \sim \pi}[Q_{\theta'_i}(s_{t+1}, a_{t+1}) - \alpha \log(\pi(a_{t+1} \mid s_{t+1}))], \tag{6}$$

where $Q_{\theta'_i}$ represents $i$-th target-critic network.

**Policy Update Iteration** The policy, parameterized by $\phi$, distills the softmax policy induced by the soft Q-function. The discrete SAC policy directly maximizes the probability of discrete actions, in contrast to the continuous SAC policy, which optimizes the two parameters of the Gaussian distribution. Then, the discrete SAC policy is updated by minimizing KL divergence between the policy distribution and the soft Q-function.

$$\pi_{\phi_{new}} = \underset{\pi_{\phi_{old}} \in \Pi}{\arg\min} D_{\mathrm{KL}}\left(\pi_{\phi_{old}}(. \mid s_t) \middle\| \frac{\exp(\frac{1}{\alpha}Q^{\pi_{\phi_{old}}}(s_t, .))}{Z^{\pi_{\phi_{old}}}(s_t)}\right). \tag{7}$$

Note that the partition function $Z^{\pi_{\phi_{old}}}(s_t)$ is a normalization term that can be ignored since it does not affect the gradient concerning the new policy. The resulting optimization objective of the policy is as follows:

$$J_\pi(\phi) = \mathbb{E}_{s_t \sim D}[\mathbb{E}_{a_t \sim \pi_\phi}[\alpha \log(\pi_\phi(a_t \mid s_t)) - Q_\theta(s_t, a_t)]]. \tag{8}$$

**Automating Entropy Adjustment** The entropy parameter temperature $\alpha$ regulates the value-entropy balance in soft Q learning. The SAC paper proposes using the temperature Lagrange term to tune the temperature $\alpha$ automatically. The following equation can be regarded as the optimization objective satisfying an entropy constraint.

$$\max_{\pi_{0:T}} \mathbb{E}_{\rho_\pi}\left[\sum_{t=0}^{T} r(s_t, a_t)\right] \quad \text{s.t.} \; \mathbb{E}_{(s_t, a_t) \sim \rho_\pi}[-\log(\pi_t(a_t \mid s_t))] \geq \mathcal{H}, \forall t, \tag{9}$$

where $\mathcal{H}$ is the desired minimum expected entropy. Optimizing the Lagrangian term $\alpha$ involves minimizing:

$$J(\alpha) = \mathbb{E}_{(a|s) \sim \pi_t}[-\alpha \log \pi_t(a_t \mid s_t) - \alpha \mathcal{H}]. \tag{10}$$

By setting a loose upper limit on the target entropy $\mathcal{H}$, SAC achieves automatic adjustment of temperature $\alpha$. Typically, the target entropy is set to $0.98 * -log(\frac{1}{dim(Actions)})$ for discrete(Christodoulou, 2019) and $-dim(Actions)$ for continuous actions(Haarnoja et al., 2018b).

## 4 Failure Modes of Vanilla Discrete SAC

We start by outlining the failure modes of the vanilla discrete SAC and then analyze under what circumstances the standard choices of vanilla discrete SAC perform poorly.

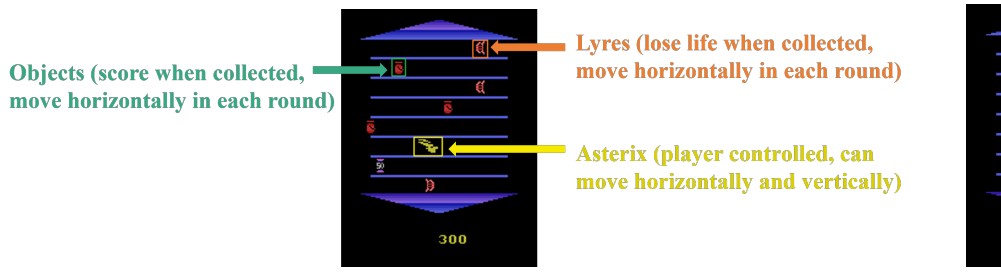

(a) Gameplay screenshot of the Atari Game Asterix

(b) Deceptive rewards in Asterix

Figure 1: Gameplay screenshot of the Atari Game Asterix, including the player-controlled Asterix (yellow box), scoring objects (green box) and life-losing lyres (orange box) that appear in rounds. Deceptive rewards appear in the early stage of game when there are only scoring objects.

## 4.1 Unstable Coupling Training

The first failure mode comes from the instability caused by fluctuations in Q function distribution and policy entropy during training. The maximum entropy mechanism in SAC effectively balances exploration and exploitation. However, due to the existence of entropy term in the soft Bellman error, and the mechanism in discrete SAC that aligns the policy with the Q function, the policy update iteration (Eq. 8) is strongly coupled with Q-learning iteration (Eq. 5).

In environments with deceptive rewards (Hong et al., 2018), the agent can gain substantial returns in the early stages of training through short-term rewards, causing the Q value of specific actions to rise rapidly and the Q function distribution to become sharper. The coupling learning paradigm of discrete SAC leads to a sharper policy distribution and, thus, a decline in entropy. Consequently, the Q learning target becomes unstable, which can, in turn, deteriorate the policy learning. As a result, the agent falls into local optima and struggles to discover alternative strategies with larger long-term payoffs. To illustrate this issue more concretely, we take the training process of discrete SAC in the Atari game Asterix as an example.

As shown in Fig. 1(a), the player controls Asterix, which can move horizontally and vertically. In each round, horizontally moving objects appear. Asterix scores points by collecting objects and loses a life when collecting a lyre. In the early stage of the game, rounds often appear where there are only scoring objects and no life-losing lyres (Fig. 1(b)), allowing the agent to score quickly by collecting objects, resulting in deceptive rewards. These rewards make the Q function sharper, thereby reducing the entropy of the policy. In Fig. 2(a), we sample a fixed set of states, and measure the variance of Q function across different actions for these states. We find that the Q function variance increases rapidly, indicating that the Q function becomes sharp quickly. Policy entropy also decreased during this period.

As the learning process continues, the policy entropy drops rapidly, and the action probabilities become deterministic quickly (Fig. 2(c)). The agent can collect objects but struggles to avoid obstacles effectively. After the policy entropy reaches its lowest point at round 2 million steps, neither the episode length (Fig. 2(d)) nor the number of steps with rewards (Fig. 2(e)) increases significantly. At the same time, the drastic change of policy entropy misleads the learning process, and thus, both Q-value and policy fall into local optimum (Fig. 2(c) and Fig. 2(b)). Since both policy and Q-value converge to the local optimum, it becomes hard for the policy to explore efficiently in the later training stage. Even the policy entropy re-rises in the later stage (Fig. 2(c))), the performance of policy does not improve anymore (Fig. 2(f)). Similar situations also occur in other Atari environments, and we provide more examples in Appendix A.6.

To better understand why this undesirable behavior occurs, we inspect the gradient of the soft Bellman object calculated by the formula 5.

$$\hat{\nabla}_\theta J_Q(\theta) = \nabla_\theta Q_\theta(a_t, s_t)(Q_\theta(s_t, a_t) - (r(s_t, a_t) + \gamma(Q_\theta(s_{t+1}, a_{t+1}) - \alpha \log(\pi_\phi(a_{t+1} \mid s_{t+1}))))). \tag{11}$$

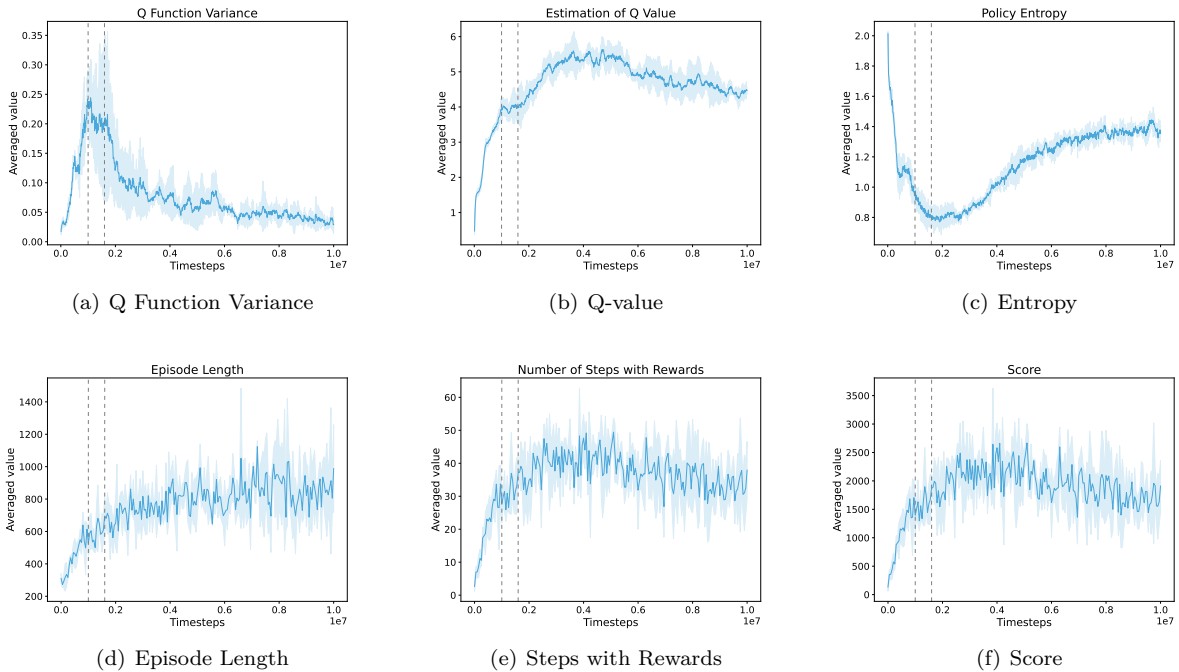

Figure 2: Measuring Q variance, estimation of Q-value, policy entropy, episode length, steps with rewards, and score on Atari Game Asterix with discrete SAC over 10 million timesteps.

As shown in Eq. 11, the improvement of $Q_\theta(s_t, a_t)$ relies on the Q-estimation of the following states and policy entropy. However, a sharper Q function causes the drastically shifting entropy, increasing the uncertainty of gradient updates and misleading the learning of the Q-network. Since the soft Q-network induces the policy, the policy can also become misleading and hurt performance. To mitigate this phenomenon, the key is to ensure the smoothness of policy entropy change to maintain stable training. In the next section, we will introduce how to constrain the policy's randomness to ensure smooth policy changes.

## 4.2 Pessimistic Exploration

The second failure mode comes from pessimistic exploration due to the double Q-learning mechanism. To address the issue of overestimation in DQN, double Q-learning was proposed. This approach mitigates the problem by employing two independent Q-networks, and using the minimum value between them as the final Q-value. The concept was initially introduced by Double DQN (Van Hasselt et al., 2016) in the discrete domain. In the continuous domain, TD3 (Fujimoto et al., 2018) and SAC (Haarnoja et al., 2018a) also adopt clipped double Q-learning to mitigate overestimation, making it a favored technique across various reinforcement learning algorithms.

Empirical results demonstrate that the clipped double Q-learning trick can boost SAC performance in continuous domains, but its impact remains unclear in discrete domains. Therefore, we need to revisit clipped double Q-learning for discrete SAC.

In our experiments, in discrete domains, we find that discrete SAC tends to suffer from underestimation bias instead of overestimation bias. This underestimation bias can cause pessimistic exploration, especially in the case of sparse reward. Here, we illustrate how the popularly used clipped double Q-learning trick causes underestimation bias and how the policy used with this trick tends to converge to suboptimal actions for discrete action spaces. Our work complements previous work with a more in-depth analysis of clipped double Q-learning. We demonstrate the existence of underestimation bias and then illustrate its impact on Atari games.

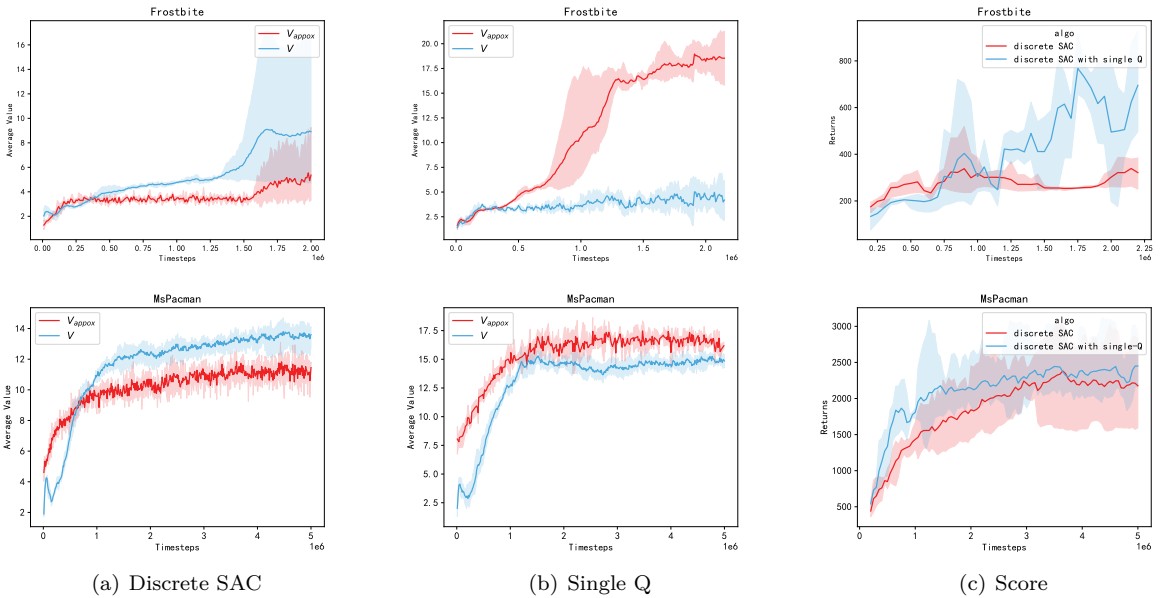

      (a) Discrete SAC             (b) Single Q             (c) Score

Figure 3: The results of Atari game Frostbite/MsPacman environment over 2/5 million time steps: a) Measuring Q-value estimates of discrete SAC; b) Measuring Q-value estimates of discrete SAC with single Q; c) Score comparison between discrete SAC and discrete SAC with single Q.

To analyze the estimated bias $\epsilon$, we introduce the mathematical expression of the soft state-value function:

$$V(s_t) = \mathbb{E}_{a_t \sim \pi}[Q(s_t, a_t) - \alpha \log(\pi(a_t \mid s_t))], \tag{12}$$

where $Q(s_t, a_t)$ represents the true Q-value. In practice, SAC uses the clipped double Q-learning trick. The learning target of soft state-value function can be written as:

$$V_{appox}(s_t) = \mathbb{E}_{a_t \sim \pi} \min_{i=1,2}[Q_{\theta'_i}(s_t, a_t) - \alpha \log(\pi(a_t \mid s_t))], \tag{13}$$

where $Q_{\theta'_i}$ represents estimation of target-critic networks parameterized by $\theta'_i$. The estimated bias for $Q'_{\theta_i}$ can be calculated as $\epsilon_i = Q_{\theta'_i}(s, a) - Q(s, a)$. On the one hand, when $\epsilon_1 > \epsilon_2 > 0$, the clipped double Q-learning trick can help mitigate overestimation error due to the *min* operation. On the other hand, when $\epsilon_1 < \epsilon_2 < 0$ or $\epsilon_1 < 0 < \epsilon_2$, the clipped double Q-learning trick will lead to underestimation (i.e., $V_{appox} < V$) and consequently result in pessimistic exploration (Pan et al., 2020; Ciosek et al., 2019).

Does this theoretical underestimate occur in practice for discrete SAC and hurt the performance? We answer this question by showing the influence of the clipped double Q-learning trick for discrete SAC in Atari games, as shown in Fig. 3. Here, we compare the true value to the estimated value. The results are averaged over three independent experiments with different random seeds. We find that, in Fig. 3(a), the approximate values are lower than the true value over time, demonstrating the underestimation bias issue. At the same time, we also run experiments for discrete SAC with a single Q (DSAC-S), which uses a single Q-value for bootstrapping instead of clipped double Q-values. As shown in Fig. 3(b), without the clipped double Q-learning trick, the estimated value of DSAC-S is higher than the true value and thus has an overestimation bias. However, in Fig. 3(c), we discover that even though DSAC-S suffers from overestimation bias, it performs much better than discrete SAC which adopts the clipped double Q-learning mechanism. This indicates that the clipped double Q-learning trick can lead to pessimistic exploration issues and hurt the agent's performance.

# 5 Improvements of SAC Failure Modes

We provide two simple alternatives, which are the surrogate objective with entropy-penalty and double average Q-learning with Q-clip, to avoid the two failure modes of discrete SAC discussed in Section 4. Combining these two modifications, we propose stable discrete SAC (SD-SAC).

## 5.1 Entropy-Penalty

The drastic change of Q function distribution and entropy affects the optimization of the Q-value. Due to the mutual coupling of the Q function and policy training in discrete SAC, we optimize policy entropy to alleviate the unstable effect on training caused by a sharp Q function distribution and a rapid drop in entropy. Simply removing the entropy term will injure the exploration ability under the framework of maximum entropy RL. An intuitive solution is to introduce an entropy penalty in the objective of policy to avoid entropy chattering. We will introduce how to incorporate the entropy penalty in the learning process for the discrete SAC algorithm.

Recall the objective of policy in discrete SAC as in Eq. 8. For a mini-batch transition data pair $(s_t, a_t, r_r, s_{t+1})$ sampled from the replay buffer, we add an extra entropy term $\mathcal{H}_{\pi_{old}}$ to the transition tuple which reflects the randomness of policy (i.e., $(s_t, a_t, r, s_{t+1}, \mathcal{H}_{\pi_{old}})$), where $\pi_{old}$ denotes the policy used for data sampling. We calculate the entropy penalty by measuring the distance between $\mathcal{H}_{\pi_{old}}$ and $\mathcal{H}_\pi$. Formally, the objective of the policy is as follows:

$$
\begin{aligned}
J_\pi(\phi) = {} & \mathbb{E}_{s_t \sim D}[\mathbb{E}_{a_t \sim \pi_\phi}[\alpha \log(\pi_\phi(a_t \mid s_t)) - Q_\theta(s_t, a_t)]] \\
& + \beta \cdot \frac{1}{2}\mathbb{E}_{s_t \sim D}([\mathbb{E}_{a_t \sim \pi_{\phi_{old}}}[-\log(\pi_{\phi_{old}})] - \mathbb{E}_{a_t \sim \pi_\phi}[-\log(\pi_\phi)])^2,
\end{aligned}
\tag{14}
$$

where $\mathbb{E}_{a_t \sim \pi_{\phi_{old}}}[-\log(\pi_{\phi_{old}})]$ represents policy entropy of $\pi_{\phi_{old}}$, $\mathbb{E}_{a_t \sim \pi_\phi}[-\log(\pi_\phi)]$ represents policy entropy of $\pi_\phi$, and $\beta$ denotes a coefficient for the penalty term and is set to 0.5 in this paper. By constraining the policy objective with this penalty term, we increase the stability of the policy learning process.

Fig. 4 shows the training curves to demonstrate how the entropy penalty mitigates the failure mode of policy drastic change. In Fig. 4(b), the entropy of discrete SAC (the blue curve) drops quickly, and the policy falls into a local optimum at the early training stage. Later, the policy stops improving and even suffers from performance deterioration, as shown in the blue curves in Fig. 4(c) and Fig. 4(d).

On the contrary, our proposed method (i.e., discrete SAC with entropy-penalty) demonstrates better stability than discrete SAC. As shown in Fig. 4(a), entropy penalty effectively constrains the sharpness of Q function, as a result, the policy changes smoothly during training (Fig. 4(b)). Consequently, compared with discrete SAC, the policy in our approach can keep improving during the whole training stage and does not suffer from a performance drop at the later training stage (the red curves in Fig. 4(c) and Fig. 4(d)).

It is worth noting that, since the instability in training mainly manifests as the existence of policy entropy term in optimization, imposing constraints in the entropy space is more effective than constraints in the policy space. Other common methods, such as the KL penalty, limit the magnitude of policy updates and impose additional restrictions on policy updates. This is proved in experiments: KL penalty (the yellow curve) cannot effectively constrain the rise in Q variance (Fig. 4(a)) and the decrease in entropy (Fig. 4(b)). Consequently, the final Q-value and score of the KL penalty are lower than those with the entropy penalty, with a difference of 12% and 23%, respectively.

The entropy-penalty term $\frac{1}{2}\mathbb{E}_{s_t \sim D}([\mathbb{E}_{a_t \sim \pi_{\phi_{old}}}[-\log(\pi_{\phi_{old}})] - \mathbb{E}_{a_t \sim \pi_\phi}[-\log(\pi_\phi)])^2$, in conjunction with the temperature $\alpha$, jointly regulates the exploration of policy. Different from other trust region methods such as KL constraint (Schulman et al., 2015) or clipping surrogate objective (Schulman et al., 2017), our method penalizes the change of action entropy between old and new policies to address policy instability during training. By adding regularization in entropy space instead of policy space, our method can mitigate the drastic changes of policy entropy while maintaining the inherent exploratory ability of discrete SAC (as shown in Fig. 4(b), the policy entropy changes smoothly. It keeps at a relatively high value to encourage exploration).

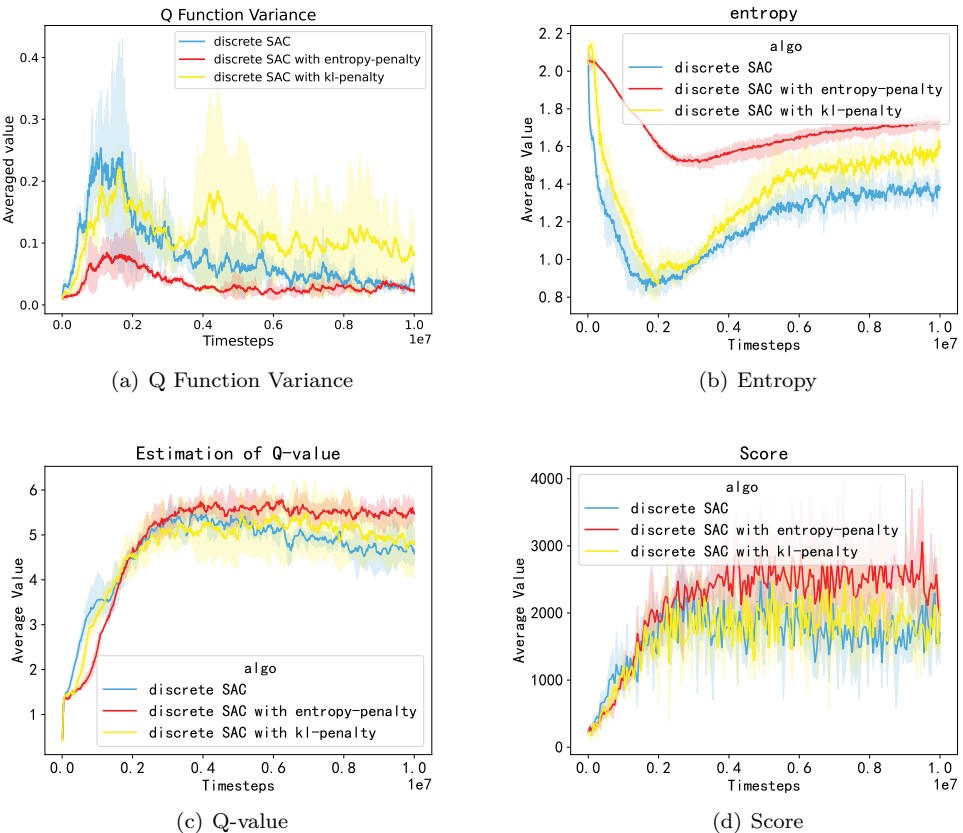

(a) Q Function Variance

(b) Entropy

(c) Q-value

(d) Score

Figure 4: Measuring Q function variance, policy action entropy, estimation of Q-value, and score on Atari game Asterix compared between discrete SAC, discrete SAC with KL-penalty and discrete SAC with entropy-penalty over 10 million time steps.

## 5.2 Double Average Q-learning with Q-clip

While several approaches(Ciosek et al., 2019; Pan et al., 2020) have been proposed to reduce underestimation bias, they are not straightforward to be applied to discrete SAC due to the use of Gaussian distribution. In this section, we introduce a novel variant of double Q-learning to mitigate the underestimation bias for discrete SAC.

In practice, discrete SAC uses clipped double q-learning with a pair of target critics $(Q_{\theta'_1}, Q_{\theta'_2})$, and the learning target of these two critics is:

$$y = r + \gamma \min_{i=1,2} Q_{\theta'_i}(s', \pi(s')). \tag{15}$$

When neural networks approximate the Q-function, there exists an unavoidable bias in the critics. Since policy is optimized concerning the low bound of double critics, for some states, we will have $Q_{\theta'_2}(s, \pi_\phi(s)) > Q_{true} > Q_{\theta'_1}(s, \pi_\phi(s))$. This is problematic because $Q_{\theta'_1}(s, \pi_\phi(s))$ will generally underestimate the true value, and this underestimated bias will be further exaggerated during the whole training phase, which results in pessimistic exploration.

To address this problem, we propose to mitigate the underestimation bias by replacing the *min* operator with *avg* operator. This results in taking the average between the two estimates, which we refer to as *double average Q-learning*:

$$y = r + \gamma \cdot \mathrm{avg}(Q_{\theta'_1}(s', \pi(s')), Q_{\theta'_2}(s', \pi(s'))). \tag{16}$$

By doing so, the other critic can mitigate the underestimated bias of the lower bound of double critics. To improve the stability of the Q-learning process, inspired by value clipping in PPO (Schulman et al., 2017), we further add a clip operator on the Bellman error to prevent drastic updates of the Q-network. The modified Bellman loss of Q-network is as follows:

$$\mathcal{L}(\theta_i) = \max\left((Q_{\theta_i} - y)^2, (Q_{\theta'_i} + \text{clip}(Q_{\theta_i} - Q_{\theta'_i}, -c, c)) - y)^2\right), \tag{17}$$

where $Q_{\theta_i}$ represents the critic network's estimate, $Q_{\theta i'}$ represents estimation of target-critic networks, and $c$ is the hyperparameter denoting the clip range. This clipping operator prevents the Q-network from performing an incentive update beyond the clip range. In this way, the Q-learning process is more robust to the abrupt change in data distribution. Combining the clipping mechanism (Eq. 17) with double average Q-learning (Eq. 16), we refer to our proposed approach as *double average Q-learning with Q-clip*.

Fig. 5 demonstrates the effectiveness of our approach. We compare the discrete SAC and various ensemble Q-estimation methods, including Randomized Ensembled Double Q-learning (REDQ) Chen et al. (2021b) and Random Ensemble Mixture (REM) Agarwal et al. (2020), with our proposed method, SD-SAC. In Fig. 5(a), the Q-value estimate of discrete SAC is underestimated than the true value. Therefore, the policy of discrete SAC suffers from pessimistic exploration and results in poor performance (purple curve in Fig. 5(e)). On the contrary, in Fig. 5(d), with double average Q-learning and Q-clip, the Q-value estimate eliminates underestimation bias and improves quickly at the early training stage. The improvement of Q-value carries over to the performance of policy. Consequently, our approach outperforms baseline discrete SAC by a large margin (Fig. 5(e)). The result also demonstrates that even though REDQ has less estimation bias in Fig. 5(b), it still suffers from underestimation bias, leading to suboptimal performance due to pessimistic exploration. Although REM addresses the underestimation issue in Fig. 5(c), the overestimation bias of REM significantly exceeds that of our proposed method, resulting in a rapid decline in performance at 8 million steps. In Fig. 5(d), we also notice that the Q-value overestimates the true value during the early training stage but finally converges to the true value after the training process. This encourages early exploration, which is consistent with the principle of optimism in the face of uncertainty (Kearns & Singh, 2002).

### 5.3 Psudocode

Finally, we provide the pseudo code for SD-SAC (i.e., Stable Discrete SAC with entropy-penalty and double average Q-learning with Q-clip), as shown in Algorithm 1.

## 6 Experiments

### 6.1 Experimental Setup

To evaluate our algorithm, we compare our SD-SAC with most related baselines, i.e., discrete SAC (Christodoulou, 2019), TES-SAC (Xu et al., 2021), Soft-DQN (Vieillard et al., 2020) and Rainbow(Hessel et al., 2018) which is widely accepted algorithm in the discrete domain. We measure their performance in 20 Atari games chosen as the same as (Christodoulou, 2019) for a fair comparison. We evaluate for 10 episodes for every 50000 steps during training, and execute 3 random seeds for each algorithm for 10 million environment steps (or 40 million frames). For the baseline implementation of discrete-SAC, we use Tianshou [1]. We find that Tianshou's implementation performs better than the original paper by Christodoulou (Christodoulou, 2019), thus we use the default hyperparameters in Tianshou on all 20 games.

We start the game with up to 30 no-op actions, similar to (Mnih et al., 2013), to provide the agent with a random starting position. To obtain summary statistics across games, following Hasselt (Van Hasselt et al., 2016), we normalize the score for each game as follows: $\text{Score}_{\text{normalized}} = \frac{\text{Score}_{\text{agent}} - \text{Score}_{\text{random}}}{\text{Score}_{\text{human}} - \text{Score}_{\text{random}}}$.

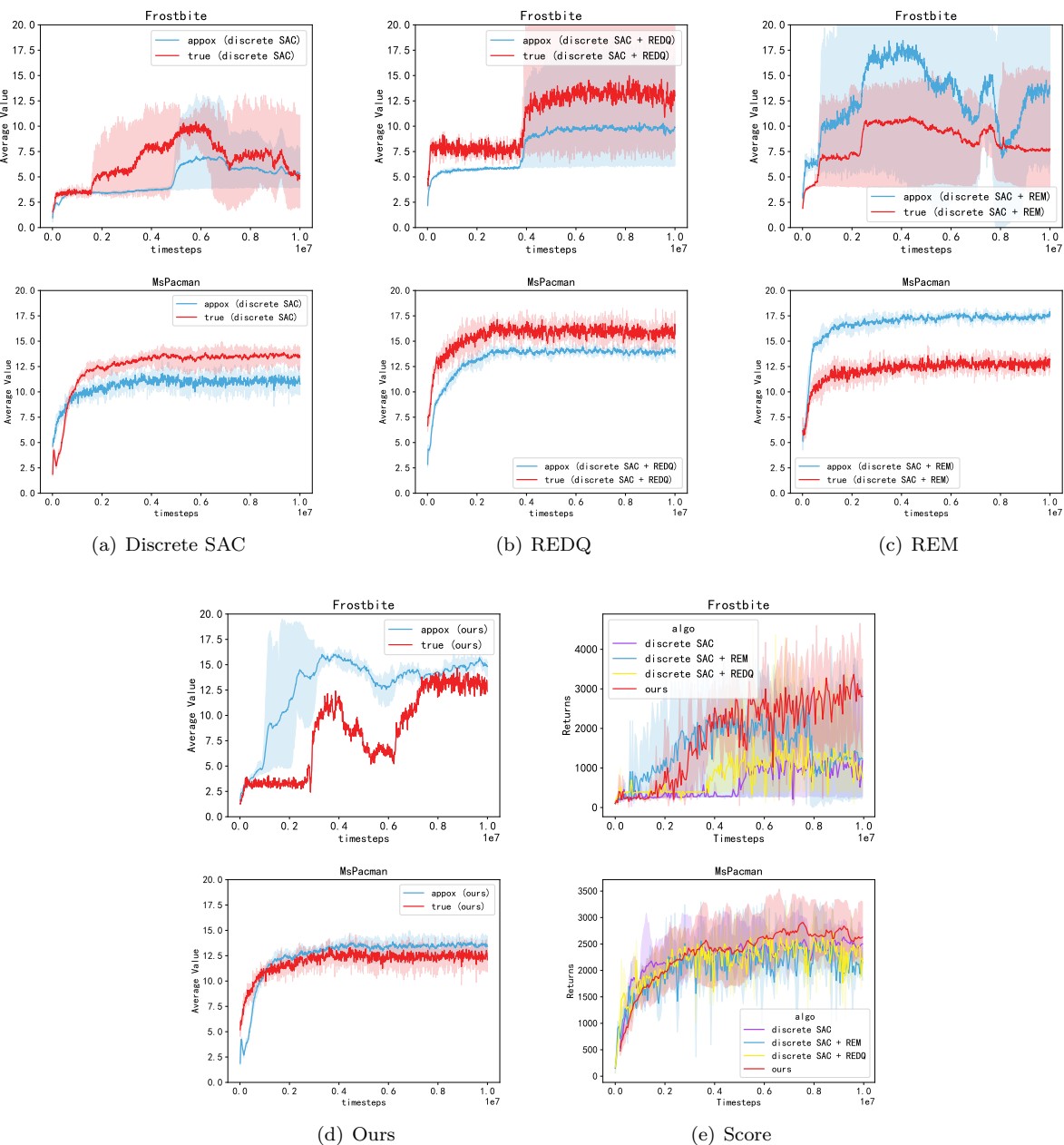

Figure 5: Measuring estimation of Q-value and score on Atari Game Frostbite/MsPacman environment compared between discrete SAC, discrete SAC with REDQ, discrete SAC with REM, and ours (SD-SAC) over 10 million steps.

Table 1: Mean and median normalized scores of discrete SAC, TES-SAC, Rainbow, Soft-DQN and our method across all 20 Atari games at 1M and 10M steps.

|  | Discrete SAC(1M) | TES-SAC(1M) | Soft-DQN(1M) | Ours(1M) | Rainbow(10M) | Discrete SAC(10M) | Soft-DQN(10M) | Ours(10M) |
|---|---|---|---|---|---|---|---|---|
| Mean | 0.5% | 3.0% | **41.7%** | 38.5% | 187.4 % | 151.4% | 199.2% | **220.0%** |
| Median | 0.4% | 2.1% | **20.0%** | 11.1% | 79.2 % | 90.8% | 107.7% | **114.1%** |

---

**Algorithm 1** SD-SAC: Stable Discrete SAC with entropy-penalty and double average Q-learning with Q-clip

---

Input: $\theta_1, \theta_2, \phi$          ▷ Initial parameters
Output: $\theta_1, \theta_2, \phi$          ▷ Optimized parameters
Hyperparameters: $\gamma, \beta, c, \tau$
Initialise $Q_{\theta_1} : S \to \mathbb{R}^{|A|}, Q_{\theta_2} : S \to \mathbb{R}^{|A|}, \pi_\phi : S \to [0,1]^{|A|}$    ▷ Initialise local networks
Initialise $Q_{\theta'_1} : S \to \mathbb{R}^{|A|}, Q_{\theta'_2} : S \to \mathbb{R}^{|A|}$    ▷ Initialise target networks
$\theta'_1 \leftarrow \theta_1, \theta'_2 \leftarrow \theta_2$     ▷ Equalise target and local network weights
$\mathcal{D} \leftarrow \emptyset$     ▷ Initialize an empty replay buffer
**for** each iteration **do**
     **for** each environment step **do**
         $a_t \sim \pi_\phi(a_t \mid s_t)$     ▷ Sample action from the policy
         $s_{t+1} \sim p(s_{t+1} \mid s_t, a_t)$     ▷ Sample transition from the environment
         $\mathcal{H}_{\pi_{old}} \sim \mathbb{E}_{a \sim \pi_\phi(\cdot | s_t)}[-\log \pi_\phi(a \mid s_t)]$    ▷ Calculate the entropy $\mathcal{H}_{\pi_{old}}$ of the current policy $\phi$
         $\mathcal{D} \leftarrow \mathcal{D} \cup \{(s_t, a_t, r(s_t, a_t), s_{t+1}, \mathcal{H}_{\pi_{old}})\}$    ▷ Store the transition in the replay buffer
     **end for**
     **for** each gradient step **do**
         $y \sim r(s_t, a_t) + \gamma \cdot \mathrm{avg}(Q_{\theta'_1}(s_{t+1}, \pi(s_{t+1})), Q_{\theta'_2}(s_{t+1}, \pi(s_{t+1})))$    ▷ Double average Q-value
estimation
         $\mathcal{L}(\theta_i) \sim \max\left((Q_{\theta_i} - y)^2, (Q_{\theta'_i} + \mathrm{clip}(Q_{\theta_i} - Q_{\theta'_i}, -c, c)) - y)^2\right)$ for $i \in \{1, 2\}$    ▷ Clip the Q-value
estimation from target critic network
         $\theta_i \leftarrow \theta_i - \lambda_Q \hat{\nabla}_{\theta_i} \mathcal{L}(\theta_i)$ for $i \in \{1, 2\}$    ▷ Update the Q-function parameters
         $\mathcal{H}_\pi \sim \mathbb{E}_{a \sim \pi_\phi(\cdot | s_t)}[-\log \pi_\phi(a \mid s_t)]$    ▷ Calculate the entropy $\mathcal{H}_\pi$ of policy $\phi$
         $J_\pi(\phi) \sim \mathbb{E}_{s_t \sim D}[\mathbb{E}_{a_t \sim \pi_\phi}[\alpha \log(\pi_\phi(a_t \mid s_t)) - Q_\theta(s_t, a_t)]] + \beta \cdot \frac{1}{2}(\mathcal{H}_{\pi_{old}} - \mathcal{H}_\pi)^2$
         $\phi \sim \phi - \lambda_\pi \hat{\nabla}_\phi J_\pi(\phi)$     ▷ Update policy weights
         $\alpha \sim \alpha - \lambda \hat{\nabla}_\alpha J(\alpha)$     ▷ Update temperature
         $Q_{\theta'_i} \leftarrow \tau Q_{\theta_i} + (1-\tau) Q_{\theta'_i}$ for $i \in \{1, 2\}$    ▷ Update target network weights
     **end for**
**end for**

---

Table 2: Raw scores across all 20 Atari games. For methods discrete SAC (1M) and TES-SAC(1M), the scores come from the corresponding paper, and the NE means the score does not exist in the original paper.

| Game | Discrete SAC (1M) | TES-SAC(1M) | Soft-DQN(1M) | Ours(1M) | Rainbow(10M) | Discrete SAC (10M) | Soft-DQN(10M) | Ours (10M) |
|---|---|---|---|---|---|---|---|---|
| Alien | 216.90 | 685.93 | 726.33 | **981.67** | 1798.33 | **2717.67** | 2018.00 | 2158.33 |
| Amidar | 7.9 | 42.07 | 130.03 | **132.97** | 394.23 | 354.77 | **438.80** | 407.20 |
| Assault | 350.0 | 337.03 | 881.97 | **1664.77** | 1802.53 | 7189.97 | **7258.87** | 6785.60 |
| Asterix | 272.0 | 378.5 | 676.67 | **733.33** | 5853.33 | 2860.00 | 3761.67 | **5993.33** |
| BattleZone | 4386.7 | 5790 | **7933.33** | 6266.67 | 24266.67 | 16850.00 | **24733.33** | 9466.67 |
| BeamRider | 432.1 | NE | 3321.60 | **3468.60** | 3310.40 | 7169.60 | 7048.20 | **10506.60** |
| Breakout | 0.7 | 2.65 | **46.17** | 11.47 | **492.93** | 29.03 | 155.83 | 60.43 |
| CrazyClimber | 3668.7 | 4.0 | **25390.00** | 20753.33 | 30286.67 | 126320.00 | 95156.67 | **140726.67** |
| Enduro | 0.8 | NE | **54.23** | 0.93 | 1517.70 | 1326.77 | 1144.07 | **2246.40** |
| Freeway | 4.4 | 13.57 | 17.70 | **20.17** | 20.13 | 15.73 | **32.30** | 20.17 |
| Frostbite | 59.4 | 81.03 | 294.33 | **347.00** | 4163.67 | 646.33 | 2959.00 | **4806.00** |
| Jamesbond | 68.3 | 31.33 | 273.33 | **368.33** | 656.67 | 1386.67 | 965.00 | **2085.00** |
| Kangaroo | 29.3 | **307.33** | 160.00 | 120.00 | 3716.67 | 2426.67 | 2703.33 | **5556.67** |
| MsPacman | 690.9 | 1408 | 1528.00 | **1639.00** | 2738.67 | **3221.33** | 2386.33 | 3175.67 |
| Pong | -20.98 | -20.84 | **20.00** | 15.53 | **20.93** | 20.37 | 20.73 | 20.37 |
| Qbert | 280.5 | 74.93 | **1400.83** | 986.67 | 15299.17 | 12946.67 | 14293.33 | **15325.83** |
| RoadRunner | 305.3 | NE | 5510.00 | **12793.33** | **45173.33** | 34043.33 | 33370.00 | 43203.33 |
| SpaceInvaders | 160.8 | NE | **488.83** | 383.50 | **1330.53** | 458.81 | 816.00 | 586.50 |
| Seaquest | 211.6 | 116.73 | 681.33 | **744.00** | 2105.33 | 1853.33 | **3438.67** | 2764.00 |
| UpNDown | 250.7 | 207.6 | **8727.33** | 8114.67 | 9110.00 | 17803.33 | **79313.00** | 63441.33 |

## 6.2 Overall Performance

Table 1 provides an overview of results and detailed results are presented in the Table 2 and Appendix A.1. Since TES-SAC is not open-sourced and our re-implement algorithm following the paper underperforms the

---

[1]https://github.com/thu-ml/tianshou

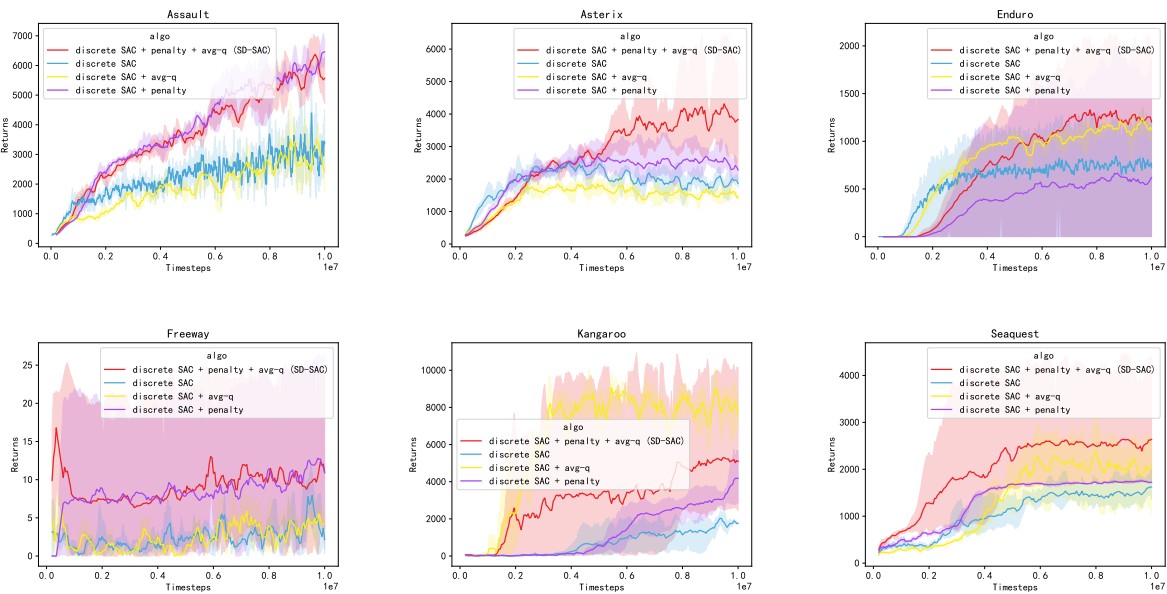

Figure 6: Scores of variant discrete SAC, which includes discrete SAC, discrete SAC with entropy-penalty, discrete SAC with double average Q learning with Q-clip,for Atari games Assault, Asterix, Enduro, Freeway, Kangaroo and Seaquest.

reported results, we adopt the normalized scores of discrete SAC and TES-SAC reported in the corresponding publication (Xu et al., 2021). When comparing our method to the discrete SAC and TES-SAC, mean normalized scores increase by 38% and 35.5%, respectively. And our method improves the median normalized scores by 10.7% and 9.0% while compared with discrete SAC and TES-SAC.

To verify the effect of a longer training process, table 1 also compares discrete SAC, Rainbow, Soft-DQN, and our method performance on 10 million steps. Compared with discrete SAC, our method has improved the normalized scores by 68.6% and 23.3% on mean and median, respectively. Additionally, our proposed method outperformed Rainbow by 32.6% on the mean and by 34.9% on the median. Better Q-estimation and steady policy updates are responsible for the performance increase in average scores. The experimental results demonstrate that benefiting from the deterministic greedy policy and entropy regularization in the evaluation step, Soft-DQN's performance improves rapidly in the early stages and achieves the best results at 1 million steps. However, due to the early convergence of the deterministic greedy policy, Soft-DQN's performance stagnates after 4 million steps, as seen in Fig. 8. Our method outperforms Soft-DQN in the final 10 million steps by 20.8% on average and 6.4% on median, due to the training stability brought by entropy penalty and the optimistic exploration altered by the double avg-Q with Q-clip.

### 6.3 Ablation Study

Fig. 6 shows the learning curves for 6 environments. Entropy-penalty (purple curve) increases performance compared to the discrete SAC in each of the six environments and even increases 2x scores in Assault. This shows that discrete SAC can perform excellently after removing unstable training. Except for Asterix, the alternative choice of clipped double Q-learning, which is double average Q learning with Q-clip (yellow curve), also shows some improvement compared to the discrete SAC in 5 environments. Additional improvements can be derived when the combination of both alternative design choices is used simultaneously.

To evaluate the influence of hyperparameter tuning, we also conducted a comprehensive hyperparameter analysis. By experimenting with different $\alpha$ and learning rate in the discrete SAC algorithm, we identify the performance upper bound of discrete SAC. The results show that, under various $\beta$ values, SD-SAC consistently outperforms this upper bound, demonstrating that entropy penalty serves as a better and more

balanced constraint. This further confirms that SD-SAC can significantly achieve a more stable training process. We present the experiment details and results in Appendix B.2.1.

## 6.4 Qualitative Analysis

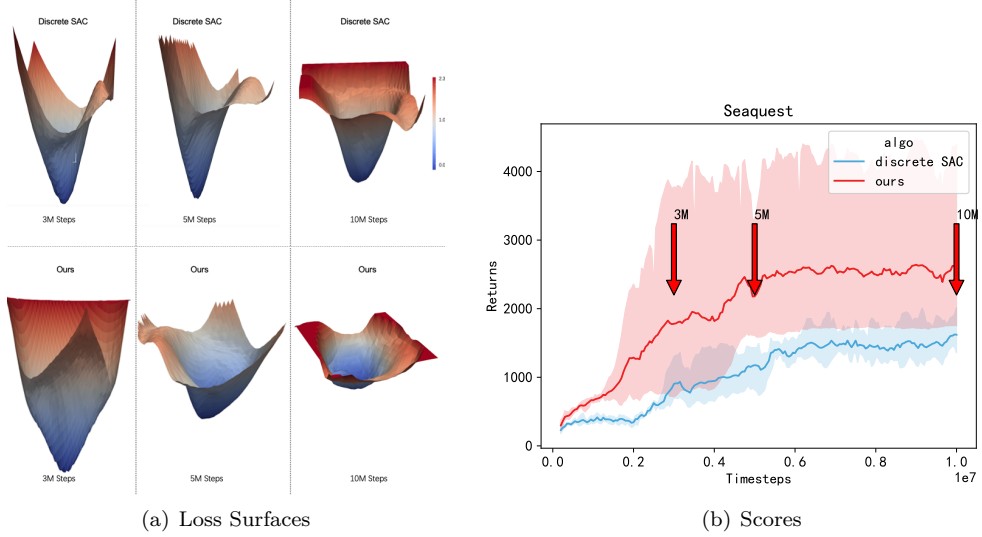

(a) Loss Surfaces                                   (b) Scores

Figure 7: The loss surfaces of discrete SAC and our method on Atari game Seaquest with trained weights at 3 million, 5 million and 10 million steps.

Fig. 7 shows loss surfaces of the discrete SAC and our method by using the visualization method proposed in (Li et al., 2018; Ota et al., 2021) with the loss of TD error of Q functions. According to the sharpness/flatness in these two sub-figures, our method has a nearly convex surface, while discrete SAC has a more complex loss surface. The surface of our method has fewer saddle points than the discrete SAC, which further shows that it can be more smoothly optimized during the training process.

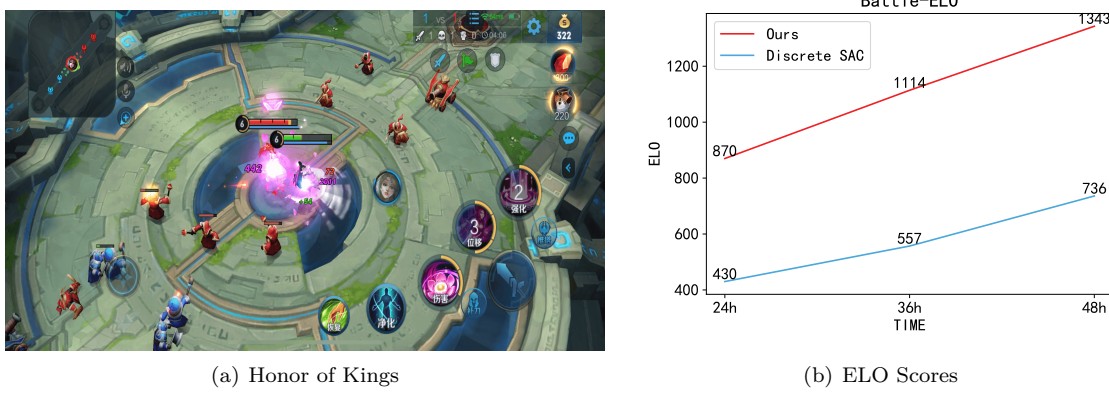

(a) Honor of Kings                                   (b) ELO Scores

Figure 8: a) A screenshot of the Honor of Kings 1v1 game. b) The ELO scores, compared with discrete SAC and our method, were tested for three snapshots of 24, 36, and 48 hours during training.

## 6.5 Case Study using Honor of Kings

We further deploy our method into Honor of Kings 1v1, a commercial game in the industry, to investigate the scale-up ability of our proposed SD-SAC algorithm.

Honor of Kings [2] is a popular MOBA (Multiplayer Online Battle Arena) game and a good testbed for RL research (Ye et al., 2020b;c;a; Chen et al., 2021a; Wei et al., 2022). The game descriptions are in (Ye et al., 2020c;a). In our experiments, we use the one-versus-one mode (1v1 solo), with both sides being the same hero: Diao Chan. The state of the game is represented by feature vectors, as reported in (Ye et al., 2020c; Wei et al., 2022). The action space is discrete, i.e., we discretize the direction of movement and skill, same to (Ye et al., 2020c;a). The goal of the game is to destroy the opponent's turrets and base crystals while protecting its own. We use the ELO rating system (Elo & Sloan, 1978), which calculate scores from the win rate, to measure the ability of two agents. A detailed introduction of the ELO system is presented in Appendix A.5.

We selected three snapshots of 24, 36, and 48 hours during the training process, resulting in 6 agents (SD-SAC-24h, SD-SAC-36h, SD-SAC-48h, DSAC-24h, DSAC-36h, DSAC-48h). We conducted 48 one-on-one matches for each agent, resulting in a total of 720 matches and thus serving as the basis of ELO calculation.

The results are shown in Fig. 8. Throughout the entire training period, our method outperforms discrete SAC by a significant margin, which indicates our method's efficiency in large-scale cases. Specifically, SD-SAC-48h achieved 35 wins, 7 draws, and 6 losses, with a win rate of 72.92%. The agent also exhibits higher skill hit rate, higher Kill/Death ratio and better turret-dashing ability.

## 7 Conclusions and Future Work

Many algorithmic design choices in reinforcement learning are limited to the regime of the chosen benchmark tasks. We highlight that soft actor-critic (SAC), that widely accepted design choices in continuous action space do not necessarily generalize to new discrete environments. We conduct failure mode analysis and obtain two main insights: 1) due to the deceptive reward, the unstable coupling update of policy and Q function will further disturb training; 2) the underestimation bias caused by double Q-learning results in the agent's pessimistic exploration and inefficient sample usage. We thereby propose two alternative design choices for SAC: entropy-penalty and double-average Q-learning with Q-clip, resulting in a new algorithm, called SD-SAC. Experiments show that our alternative design choices increase the training stability and Q-value estimation accuracy, which ultimately improves overall performance. In addition, we also apply our method to the large-scale MOBA game Honor of Kings 1v1 to show the scalability of our optimizations.

Finally, the success obscures certain flaws, one of which is that our improved discrete SAC still performs poorly in instances involving long-term decision-making. One possible reason is that SAC can not accurately estimate the future only by rewarding the current frame. In order to accomplish long-term choices with SAC, our next study will concentrate on improving the usage of the incentive signal across the whole episode.

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

# A    Additional Details and Experiment Results

## A.1    Detailed Experiment Results on 20 Atari Game Environments

In Figure  8, we present the learning curves of all 20 experiments.

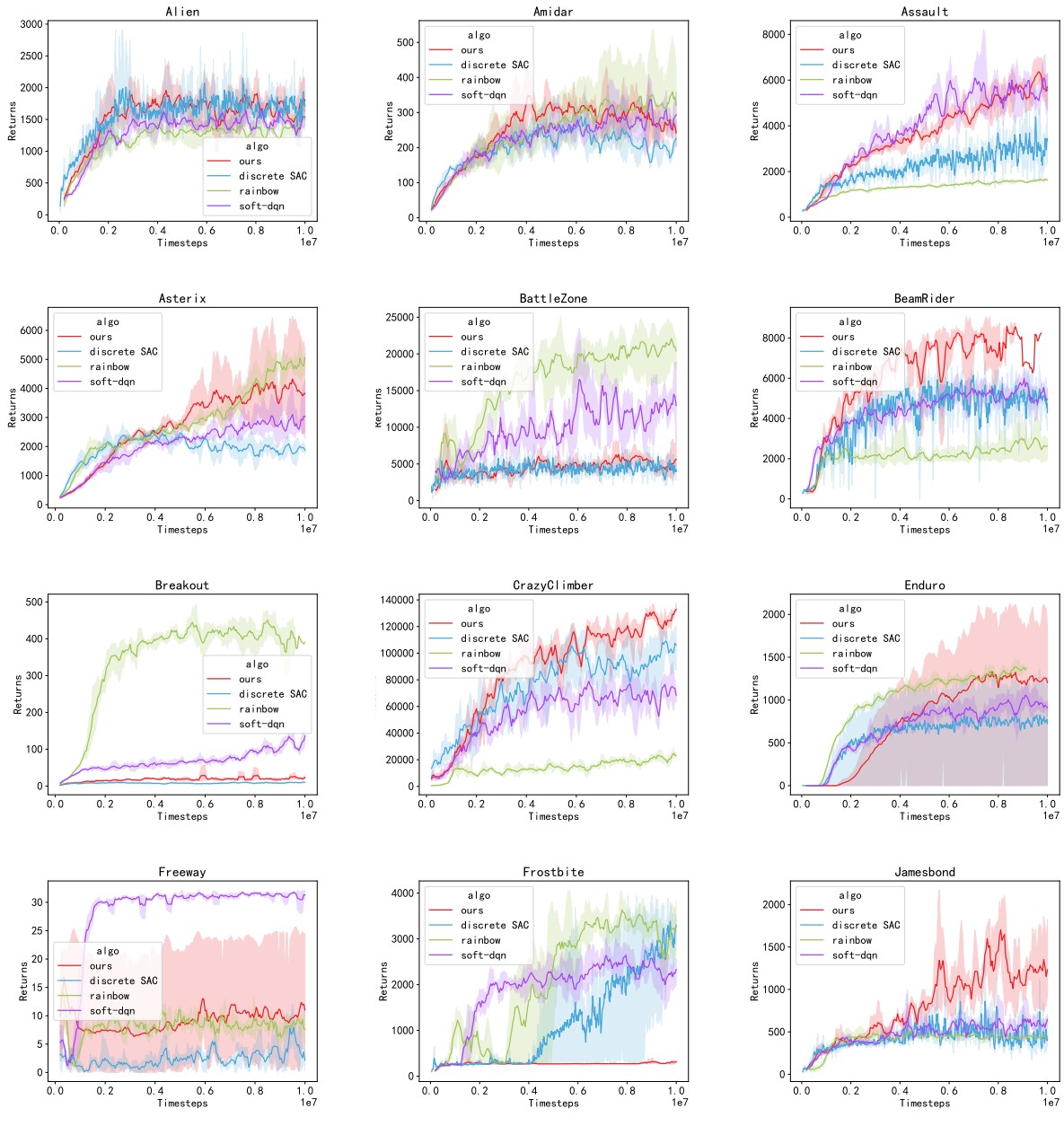

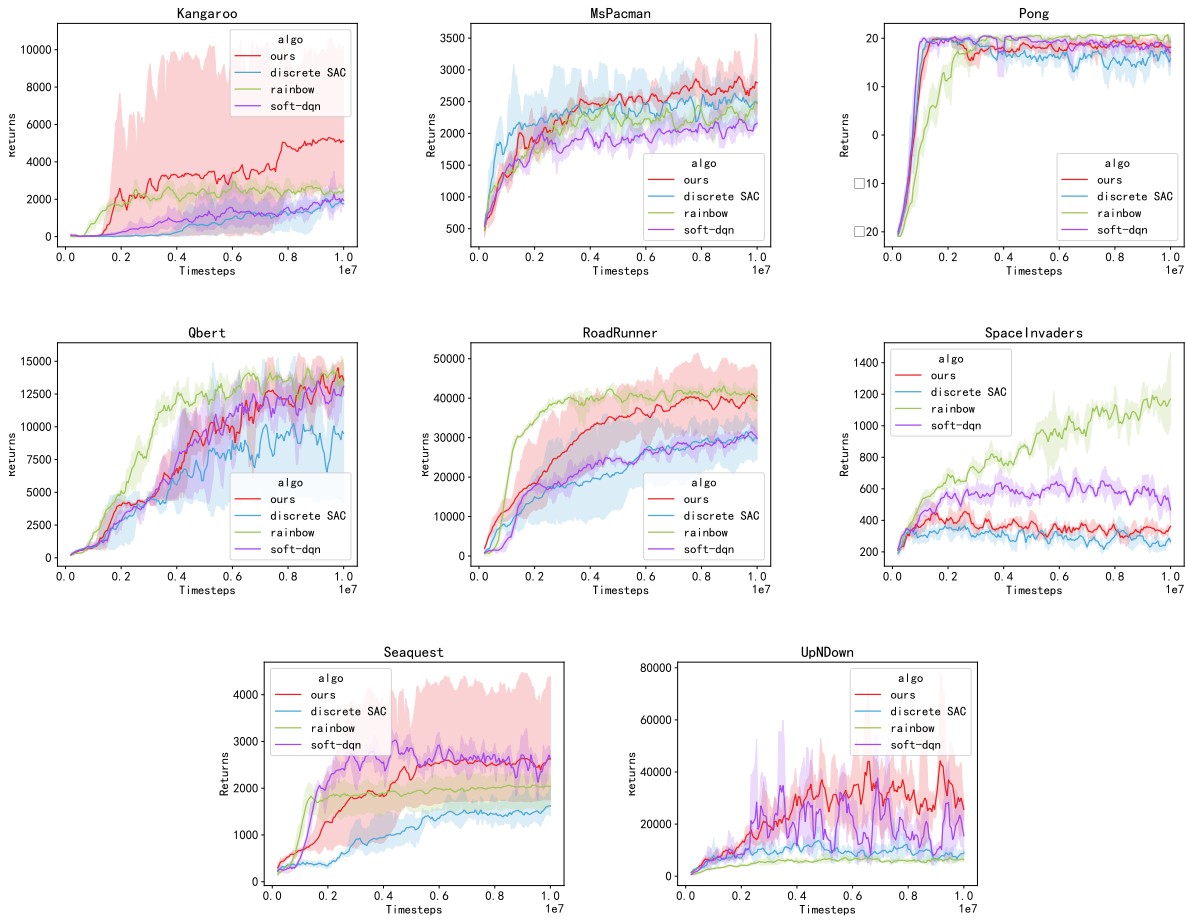

Figure 8: Learning curves for discrete SAC, Rainbow, Soft-DQN, and ours, for each game. Every curve is smoothed with a moving average of 10 to improve readability.

## A.2 Further Comparison of SD-SAC and DSAC

Beyond the comparison of Figure 4, we further measure SD-SAC and DSAC in terms of episode length and number of steps with rewards in Figure 9. After $2e6$ steps, the algorithm with entropy penalty demonstrates significant longer episode lengths and more reward steps compared to discrete SAC. This indicates that the entropy penalty helps the agent learn both scoring and avoidance skills, leading to continued performance improvement.

## A.3 Result Curves of Individual Runs

For clearer demonstration, we show the results of individual run curves for figures during our major analysis in Section 4 and 5.

### A.3.1 Comparison Between True Q Values and Estimate Q Values

We plot Figure 3 by individual runs in Figure 10. The results show that for each individual seed, discrete SAC consistently suffers from an underestimation problem, while using a single Q leads to an overestimation issue.

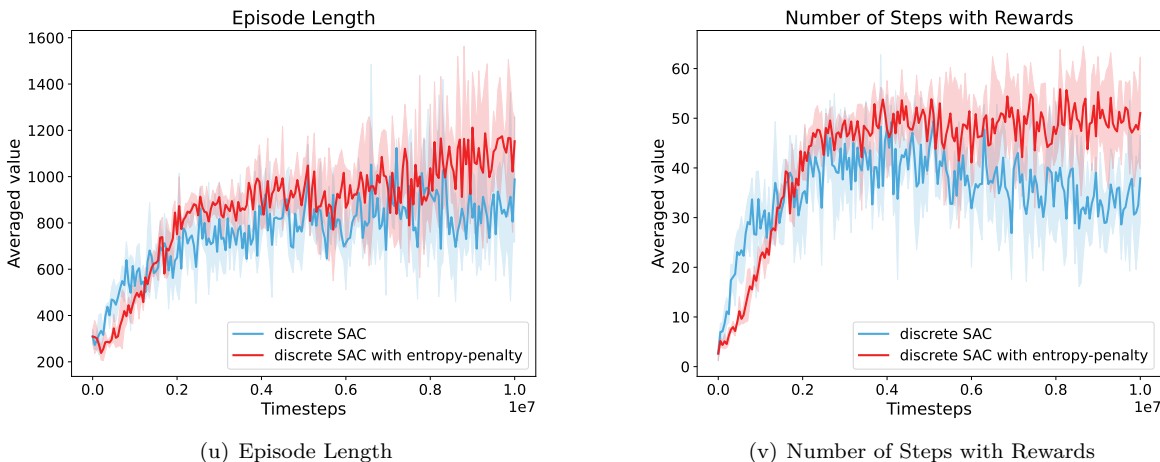

(u) Episode Length  (v) Number of Steps with Rewards

Figure 9: Comparison of SD-SAC and DSAC in terms of episode length and number of steps with rewards in Asterix.

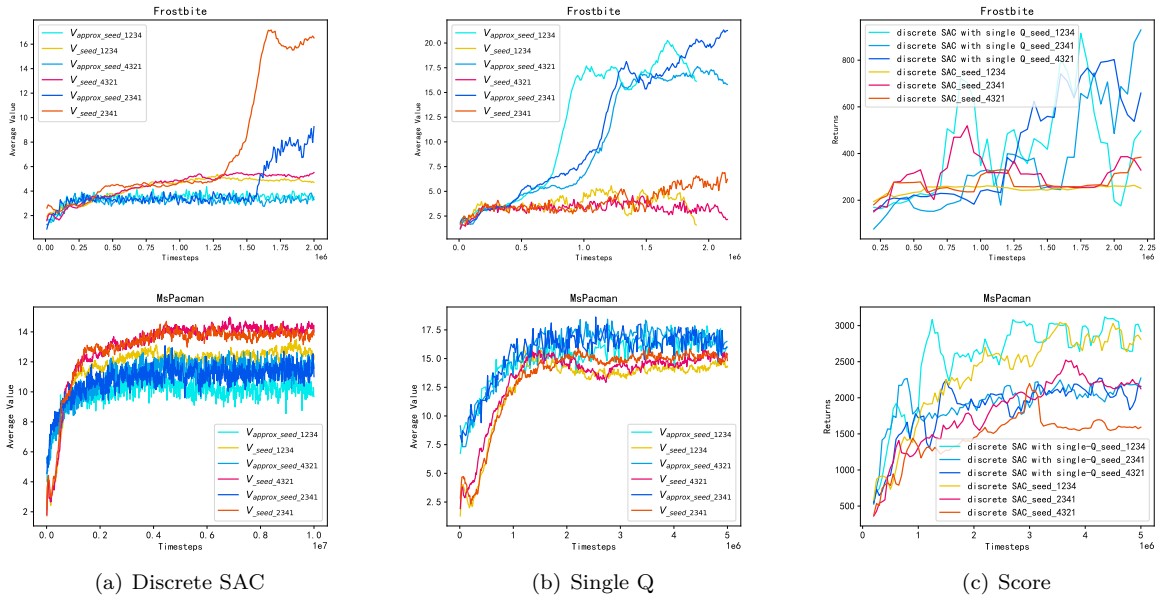

(a) Discrete SAC  (b) Single Q  (c) Score

Figure 10: The results of Atari game Frostbite/MsPacman environment over 2/5 million time steps: a) Measuring Q-value estimates of discrete SAC; b) Measuring Q-value estimates of discrete SAC with single Q; c) Score comparison between discrete SAC and discrete SAC with single Q.

### A.3.2   Comparison Between Different Policy Constraints

We provide Figure 4 by individual runs in Figure 11. The results show that entropy penalty enables a stable training and better performance across all seeds.

### A.3.3   Comparison Between Different Q-value Estimation Methods

We provide Figure 5 by individual runs in Figure 12. Our approach demonstrates effectiveness in alleviating underestimation and reduce bias in all individual runs.

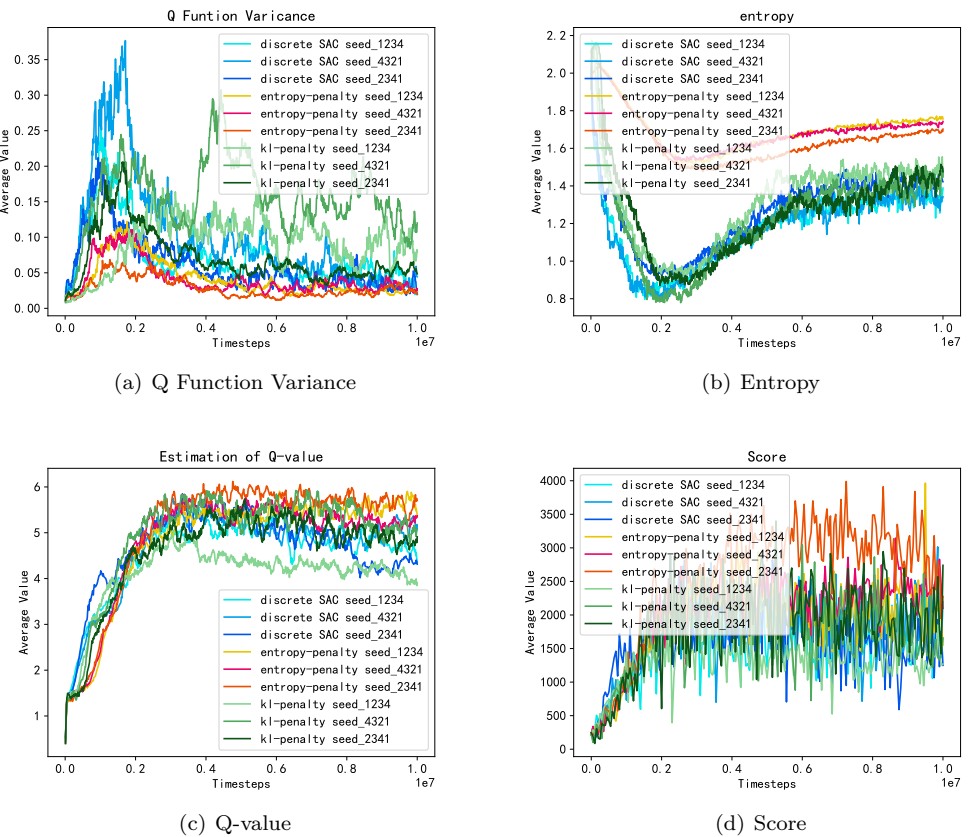

(a) Q Function Variance

(b) Entropy

(c) Q-value

(d) Score

Figure 11: Measuring Q function variance, policy action entropy, estimation of Q-value, and score on Atari game Asterix compared between discrete SAC, discrete SAC with KL-penalty and discrete SAC with entropy-penalty over 10 million time steps.

Table 3: Hyperparameter for Discrete SAC and SD-SAC

| Hyperparameter | Discrete SAC | SD-SAC |
|---|---|---|
| learning rate | $10^{-5}$ | $10^{-5}$ |
| optimizer | Adam | Adam |
| mini-batch size | 64 | 64 |
| discount ($\gamma$) | 0.99 | 0.99 |
| buffer size | $10^5$ | $10^5$ |
| hidden layers | 2 | 2 |
| hidden units per layer | 512 | 512 |
| target smoothing coefficient ($\tau$) | 0.005 | 0.005 |
| Learning iterations per round | 1 | 1 |
| alpha | 0.05 | 0.05 |
| n-step | 3 | 3 |
| $\beta$ | False | 0.5 |
| $c$ | False | 0.5 |

## A.4 Hyperparameter Used in SD-SAC

Please refer to Table 3.

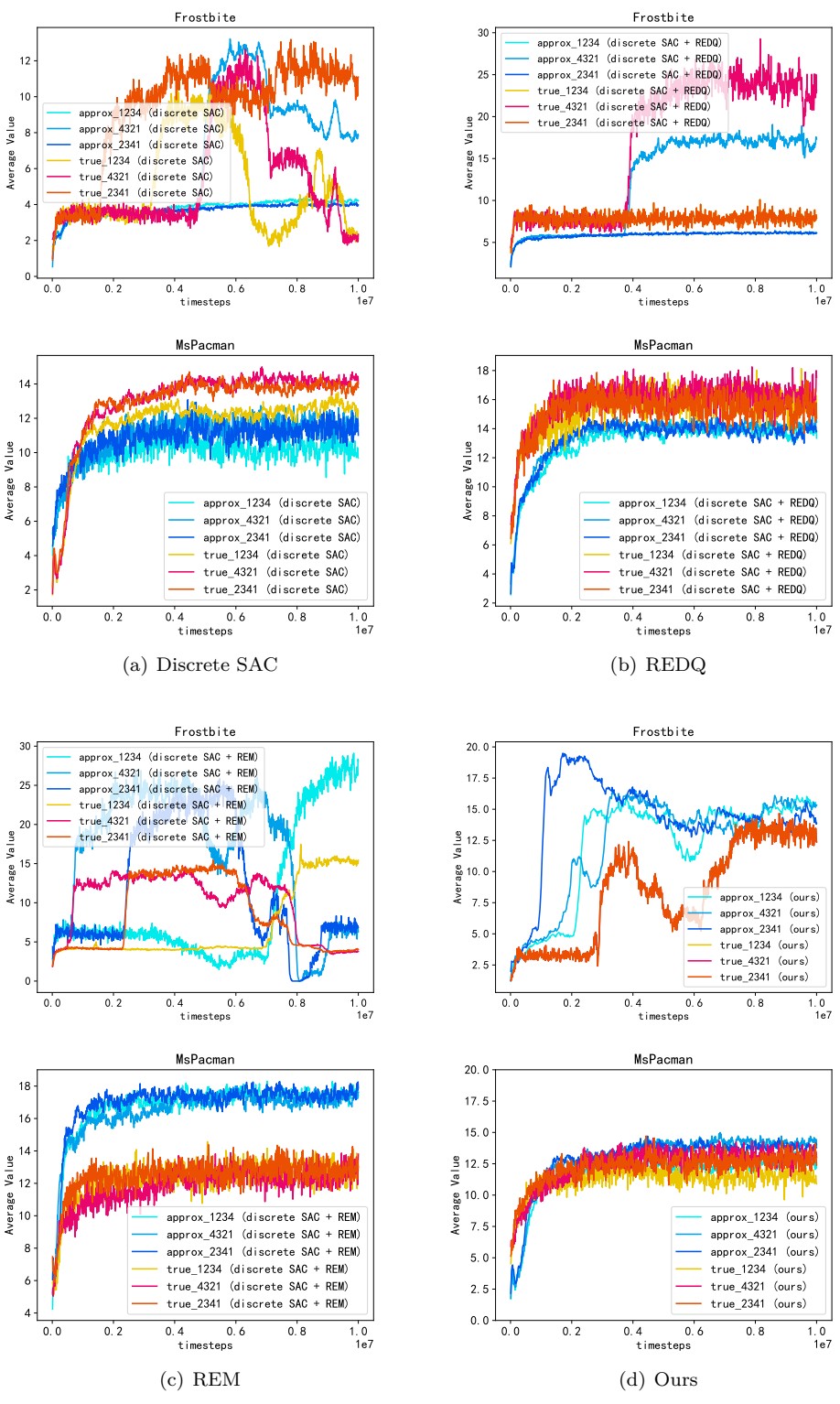

Figure 12: Measuring estimation of Q-value on Atari Game Frostbite/MsPacman environment compared between discrete SAC, discrete SAC with REDQ, discrete SAC with REM, and SD-SAC (discrete SAC with double average Q-learning with Q-clip) over 10 million steps.

### A.5 Introduction of the ELO System

The ELO rating system (Elo & Sloan, 1978) is a widely-used mechanism for assessing the relative skill levels of players or agents, commonly applied in chess, competitive games, and other adversarial environments. In our study, the ELO ratings of agents are calculated through the following process:

1. Each agent is assigned an initial ELO rating $R_{base}$.

2. Before agent A competes against agent B, the expected score for each agent is calculated based on their current ELO ratings: $E_A = \frac{1}{1+10^{(R_B-R_A)/R_{base}}}$

3. The ELO ratings are updated based on the outcome of the match between A and B, where $S_A$ is the actual score and $K$ is a constant: $R'_A = R_A + K \cdot (S_A - E_A)$

4. Through multiple matches among various agents, the ELO ratings are adjusted according to the results of these matches. The final ELO ratings reflect each agent's relative strength compared to the others.

### A.6 Other Atari Environments for Unstable Coupling Training of Discrete SAC

We conduct cross-validation in other Atari environments, as presented in Fig. 14-16 . The result show that in other environments with deceptive rewards, the rapid decrease in policy entropy due to large Q variance similarly affects training.

#### A.6.1 Games with Deceptive Rewards

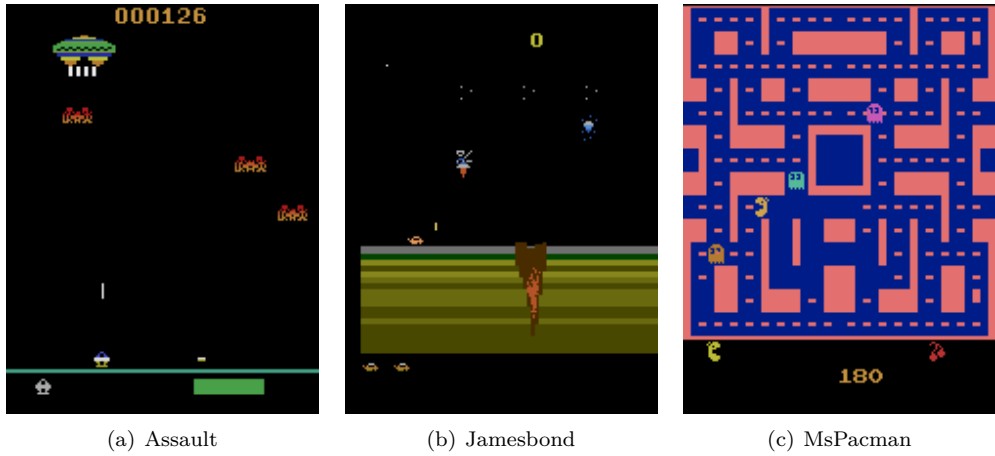

(a) Assault    (b) Jamesbond    (c) MsPacman

Figure 13: Three examples of Atari game environments with deceptive rewards.

We take the Atari games Assault, Jamesbond and MsPacman as examples to further illustrate the manifestation and impact of deceptive rewards in game environments. (Fig. 13)

In the game Assault, a mothership releases different kinds of aliens. They move along the screen, with the bottom-most alien firing various types of weapons. The player controls a cannon that can shoot bullets horizontally or vertically to attack the aliens and fireballs they shot. Hitting an alien scores points, while being hit or cannon overheating results in a loss of life.

In Jamesbond, the player controls a craft that needs to complete various mission to achieve final victory. In the first mission, the player must navigate through a desert with craters, acoid overhead satellite scans and helicopter bombings, and score points by hitting diamonds through fixed-angle shooting.

As for MsPacman, the player controls a Pacman, who scores points by eating dots in a maze while avoiding floating ghosts. When Pacman eats an energy pill, she can attack the ghosts to gain higher scores.

In all three environments, the agent can quickly gain deceptive rewards through short-term payoffs. For Assault and Jamesbond, all points come from shooting actions that hit specific targets, while avoiding obstacles can prevent the loss of life but does not bring clear rewards. Thus, agents often excel at shooting but struggle with dodging. In MsPacman, the numerous dots in the maze provide many rewards for the agent's movement. As a result, the agent finds it difficult to learn advances strategies such as avoiding ghosts and picking up energy pills to attack ghosts. The presence of deceptive rewards leads to the training process stuck in local optima, making it challenging to explore better, long-term strategies.

### A.6.2  Plots of Training Process

We present the training process in the three aforementioned environments with deceptive rewards in Fig. 14-16. It can be observed that in each case, deceptive rewards cause a rapid increase in Q variance and a decrease in policy entropy, leading the training process to fall into local optima.

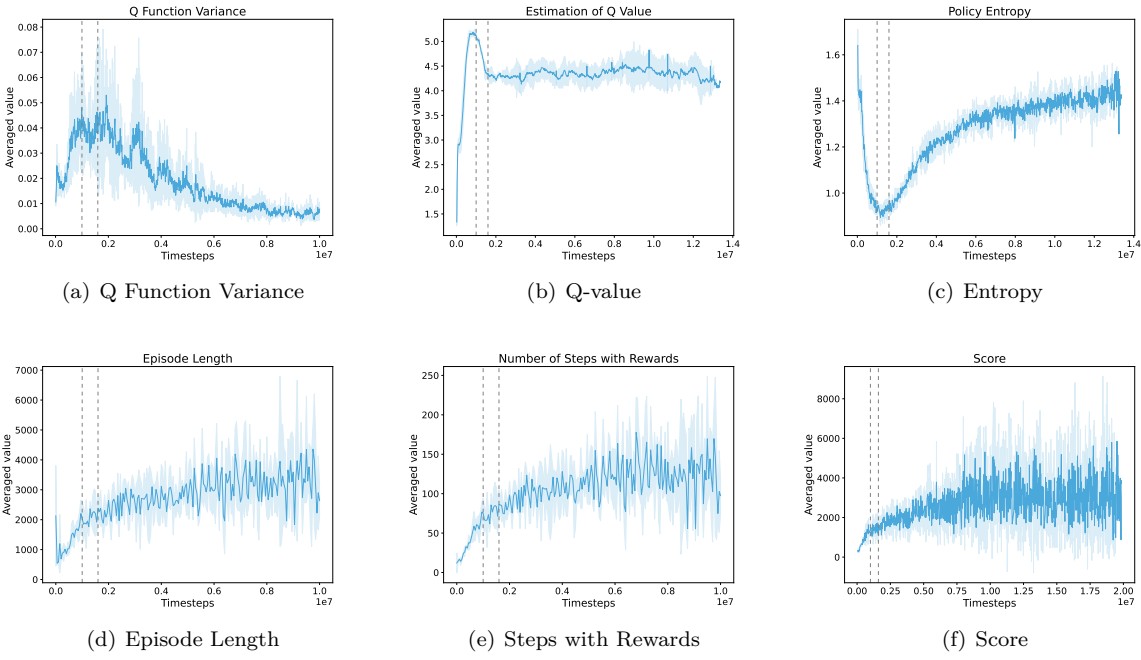

Figure 14: Plots of Q function variance, estimation of Q-value, policy action entropy, episode length, number of steps with rewards and score on Atari Game Assault environment with discrete SAC over 10 million time steps.

### A.6.3  Comparison of Different Algorithms on Additional Atari Environments

Here in Figure 17 we provide comparative performance curves of DSAC, DSAC with entropy penalty, and DSAC with KL penalty in three additional Atari game environments: Assault, Jamesbond, and MsPacman. As shown in the results, the entropy penalty consistently offers the best early-stage regulation of entropy changes across all three environments. This regulation helps prevent the agent from falling into local optimum during the learning process, thereby improving the final score performance.

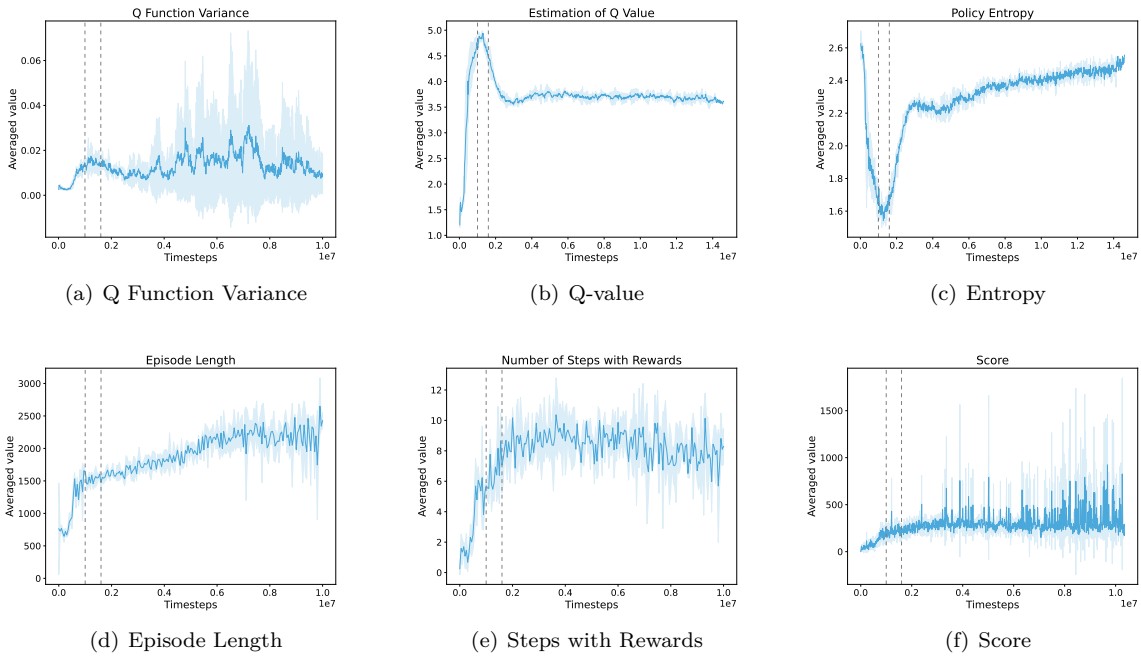

Figure 15: Plots of Q function variance, estimation of Q-value, policy action entropy, episode length, number of steps with rewards and score on Atari Game Jamesbond environment with discrete SAC over 10 million time steps.

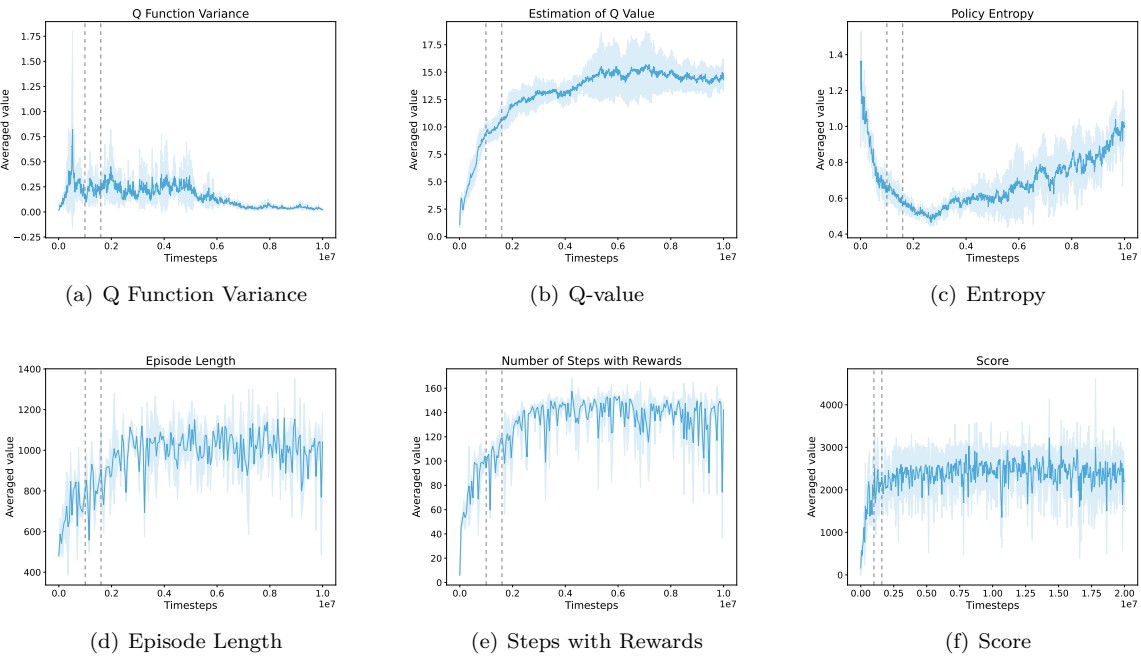

Figure 16: Plots of Q function variance, estimation of Q-value, policy action entropy, episode length, number of steps with rewards and score on Atari Game MsPacman environment with discrete SAC over 10 million time steps.

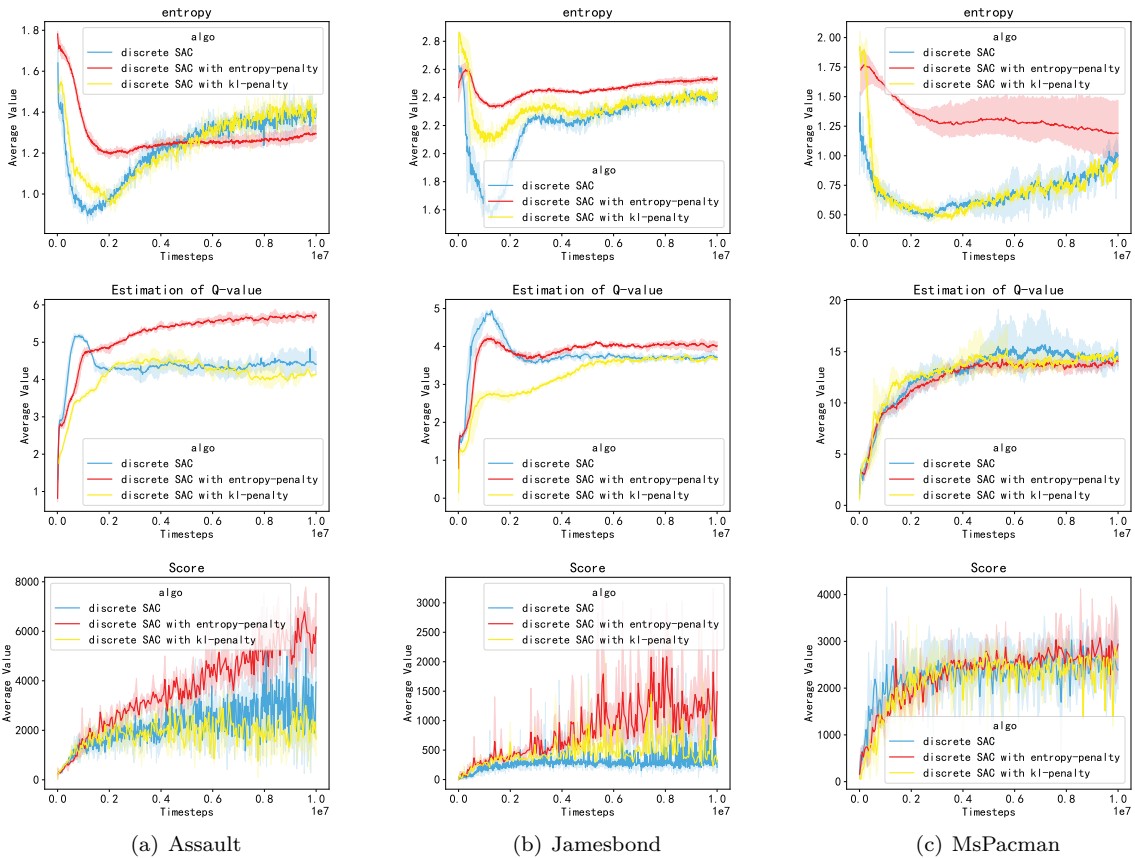

Figure 17: Measuring Q function variance, policy action entropy, estimation of Q-value, and score on Atari game Assault, Jamesbond and MsPacman, comparing between discrete SAC, discrete SAC with KL-penalty and discrete SAC with entropy-penalty over 10 million time steps.

## B  Further Analysis

### B.1  SAC Training Pattern on MuJoCo

We only observe the failure modes in discrete SAC. The reason SAC does not exhibit these failure modes in continuous environments is twofold. First, SAC employs the reparameterization trick, fitting actions with a Gaussian distribution, allowing it to adapt to deceptive rewards without sacrificing policy diversity. Second, in continuous environments, actions that deviate slightly from the best response may have minimal impact on the outcome, whereas in discrete settings, different actions can have entirely distinct meanings. Therefore, our analysis primarily focuses on the challenges SAC faces in discrete environments.

To validate this point, we test SAC on three tasks of the MuJoCo environment. Results in Figure 18 indicate that in MuJoCo, the SAC algorithm does not encounter local optimum issues; policy entropy changes are minimal and gradual, while the scores stadily increase. This suggests that SAC does not face the problems described in the paper when applied to continuous tasks.

### B.2  Hyperparameter Analysis

### B.2.1  Different Hyperparameter Choices of DSAC

Our design method incorporates two hyperparameters, i.e., entropy-penalty coefficient $\beta$ and Q-clip range $c$. Fig. 19 compares various entropy-penalty coefficient $\beta$ and Q-clip range $c$ values. The constraint proportion

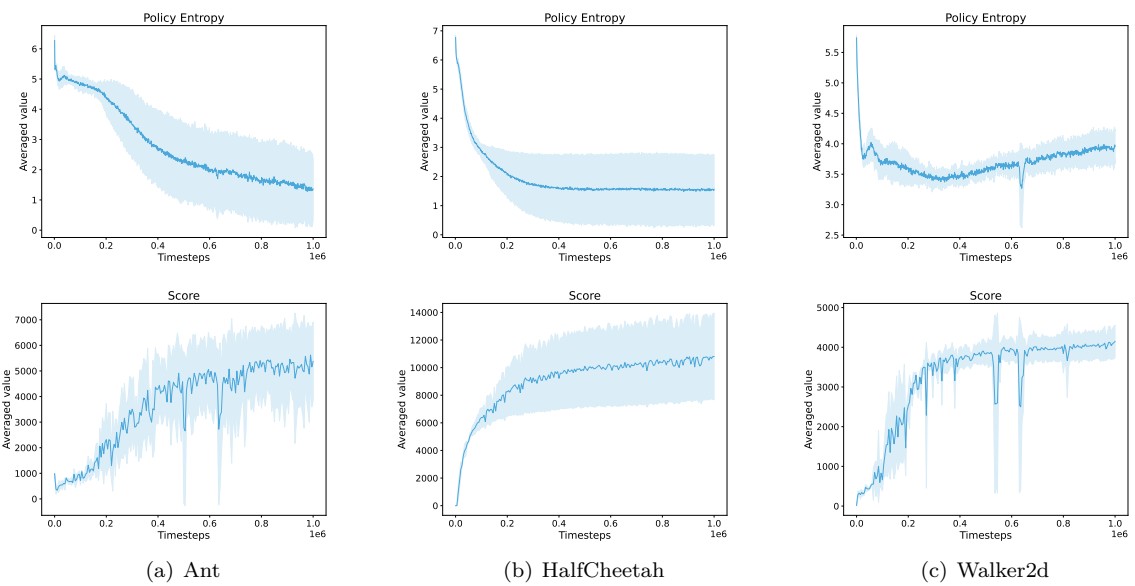

Figure 18: The results of SAC in the MuJoCo environment.

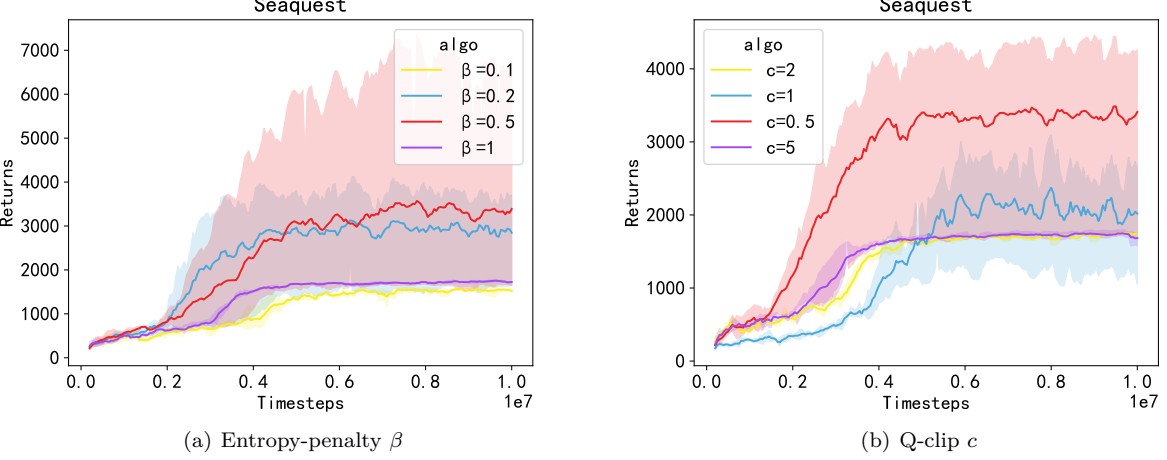

Figure 19: Scores on Seaquest: a) variants entropy-penalty coefficient $\beta$ with 0.1, 0.2, 0.5 and 1. b) variants Q-clip $c$ with 0.5, 1, 2 and 5.

of policy change is determined by the entropy-penalty coefficient $\beta$. Intuitively, an excessive penalty term will lead to policy under-optimization. We experiment with different $\beta$ in $\{0.1, 0.2, 0.5, 1\}$. We find that $\beta = 0.5$ can effectively limit entropy randomness while improving performance. The Q-clip constrains different ranges of Q value range $c$, and experiments with different ranges $c$ in $\{0.5, 1, 2, 5\}$ show that 0.5 is a reasonable constraint value.

### B.2.2 Different Choices of Clip Ratio

In Figure 20, we compare the clip ratio and final scores of different $c$ in our Q-clip.

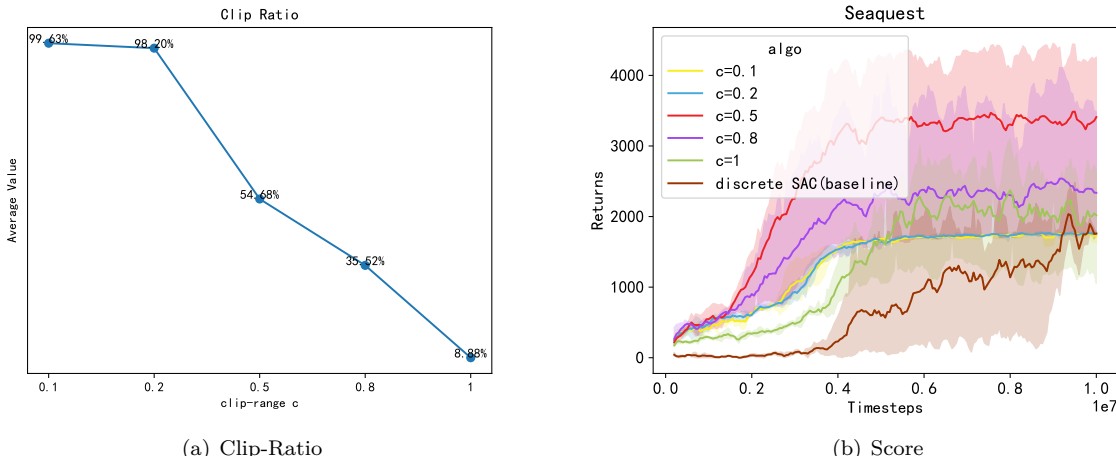

(a) Clip-Ratio

(b) Score

Figure 20: Measuring clip-ratio and score on Atari Game Seaquest environment with our method over 10 million time steps using variants Q-clip $c$ with 0.1, 0.2, 0.5, 0.8 and 1.0 .

### B.2.3 Various Learning Rates for Discrete SAC

We introduce various learning rates for experiments on Asterix using vanilla discrete SAC in Fig. 21. An excessively high learning rate leads to early convergence of entropy, while an excessively low learning rate results in insufficient optimization. The experiments show that the entropy instability issue of discrete SAC is not caused by inappropriate learning rate settings.

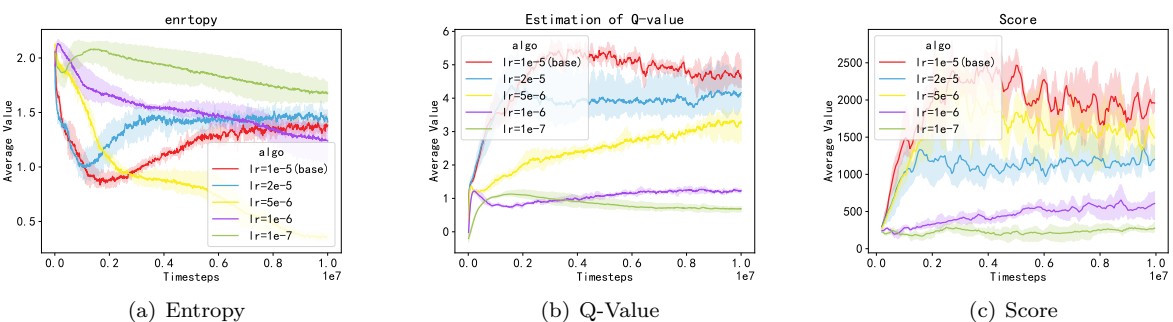

(a) Entropy

(b) Q-Value

(c) Score

Figure 21: Measuring policy action entropy, estimation of Q-value and score on Atari Game Asterix environment with discrete SAC over 10 million time steps using different learning rates.

### B.2.4 Different Choices of Temperature $\alpha$ in Discrete SAC

In Figure 22(a), we compare scores on Asterix by discrete SAC using variants $\alpha$ with 0.01, 0.025, 0.05, 0.075 and 0.1 over 10 million time steps.

### B.2.5 Comparison Across SD-SAC and DSAC

We first determine discrete SAC's best combination of $\alpha$ (Fig. 22(a)) and learning rate (Fig. 21(a)). Then, we compare the scores between SD-SAC with different $\beta$, and DSAC with this combination ($\alpha = 0.5$; lr = 1e-5) in Figure 22(b). The result shows that across various $\beta$ values, SD-SAC consistently outperforms DSAC. This indicates that the entropy penalty is a better and more balanced constraint than merely limiting the extent of policy updates.

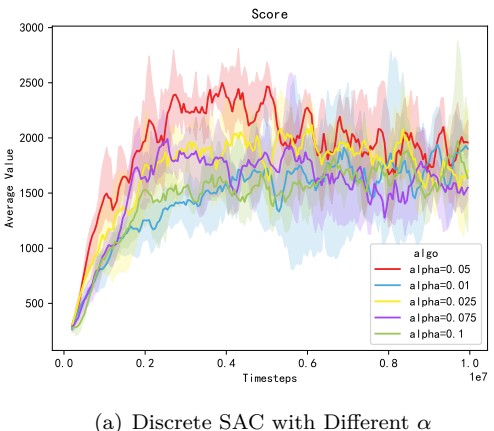 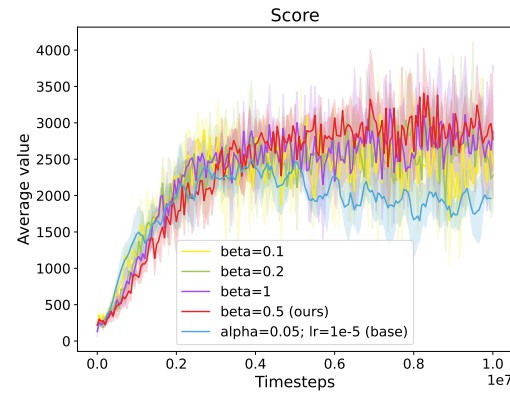

(a) Discrete SAC with Different $\alpha$

(b) SD-SAC with Various $\beta$, and DSAC with the Best $\alpha$ and Learning Rate Combination.

Figure 22: Comparison of different hyperparameters, including $\alpha$, learning rate and $\beta$.

## B.3 Computation Overhead

We test the computational speed on a machine equipped with an Intel(R) Xeon(R) Platinum 8255C CPU @ 2.50GHz with 24 cores and a single Tesla T4 GPU. The unit "it/s" represents the number of steps interacting with the environment per second. Detailed data are shown in the Table 4 below. The results demonstrate that our method has a 10.86% reduction(265.41->236.58) in speed compared to the vanilla discrete SAC, while maintaining the same parameter size.

Table 4: Computational speed our method and discrete SAC.

| algorithm | speed |
|---|---|
| discrete SAC | 265.41it/s |
| discrete SAC + entropy-penalty | 246.83it/s(-18.58) |
| discrete SAC + avg-q + q-clip | 250.27it/s(-15.14) |
| SD-SAC | 236.58it/s(-28.83) |

## B.4 Cosine Similarity Comparison

We visualize the changes in cosine similarity between adjacent states before and after incorporating the entropy penalty in the DSAC algorithm in Figure 23. The results indicate that, following the addition of the entropy penalty, state transitions exhibit smaller and more stable changes. This observation further substantiates that the entropy penalty contributes to more stable policy updates, thereby enhancing the overall performance of the algorithm.

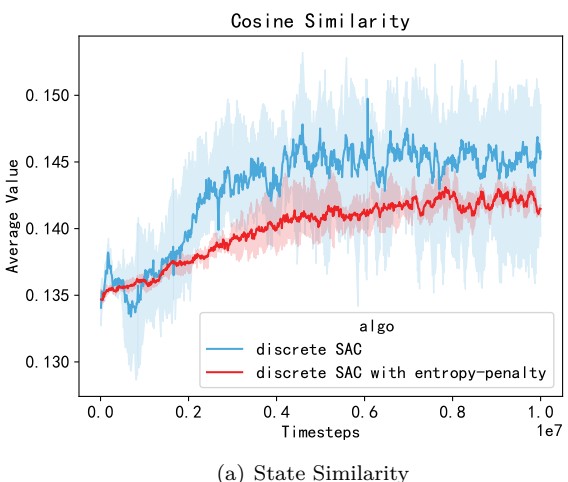

(a) State Similarity

Figure 23: Measuring cosine similarity of states on Atari game Asterix compared between discrete SAC and discrete SAC with entropy-penalty over 10 million time steps.

