# OpenReview forum: "Revisiting Discrete Soft Actor-Critic"
_TMLR — Accepted by TMLR_

### Review · Reviewer_vJSy · 2024-08-23

**Summary Of Contributions:**

This paper aims to improve the performance of the soft actor critic (SAC) approach in discrete action tasks. The authors propose some hypotheses about the issues holding back the discrete SAC algorithm and present experimental metrics as part of this discussion. They then suggest three algorithmic modifications: an entropy change penalty, averaging two critic networks, and clipping critic network predictions during critic training so they are not too far from target critic network predictions. The paper presents experimental results in Atari games and ELO scores for an application of their method in a a multi-player online battle arena game.

**Audience:**

Yes

**Broader Impact Concerns:**

I am not familiar with the game Honor of Kings so perhaps the community and stakeholders have ways to mitigate cheating with autonomous agents but a possible ethical concern is that people could use released code or re-implement methods from this paper to build policies for Honor and Kings, passing the play of those policies off as themselves.

**Claims And Evidence:**

No

**Requested Changes:**

A less critic change that would strengthen the work is to perform more repetitions to ensure that observations are more signal than noise.

### Critical Changes

1. Improve the analysis of failure modes and explain why they are specific to discrete SAC.
2. Discuss and analyze the relationship between $\alpha$, the $\alpha$ learning rate, and $\beta$. Show why the entropy change loss and $\beta$ are indeed necessary.
3. Improve plots showing values from individual runs, comparing median runs, and analyze variability more transparently by comparing the min, max, and median runs.
4. Address the optimization difficulties that Q-clipping introduces, either by modifying the approach to be sound or by showing thoroughly that the lack of soundness is necessary.
5. Improve the experiments section to clarify the points I mentioned in the previous section.
6. Address questions and concerns about the Honor of Kings case study.
7. Proofread the paper and edit wording. There are a number of awkward wordings in the paper where it is difficult to understand what is being said.

**Strengths And Weaknesses:**

The paper does not clearly explain how the failure modes of discrete SAC are specific to discrete SAC. Section 4.1 also fails to provide convincing evidence that "unstable coupling training" (awkward wording, by the way) is indeed a cause of discrete SAC's poor performance in some cases. Issues with the case study in Section 4.1:
1. In the Asterix example, how are the rewards "deceptive" and why would quickly learning to pick up all items from the early rounds be a problem? If the policy learns to pick up all items and applies this strategy in later rounds, it will quickly get data showing that collecting a lyre is actually bad and will ideally adapt to this new data quickly. This behavior is the usual progression for optimistic exploration, so why would it be a problem?

    The data in Figure 2 also does not seem to match the analysis in the main text. The episode length increase ~40% after two million steps and the number of steps with rewards increases ~27%. Both of these increases mostly happen within the next 3 million steps. These increases are both substantial compared to the shown variability, which directly contradicts the main text.
2. The text notes that the policies become deterministic "quickly" (I suppose the authors mean within two million steps). Since the authors have chosen the minimum entropy and the learning rate for $\alpha$ tuning, is this not simply a signal that the minimum entropy is too low or the $\alpha$ learning rate is poorly tuned for this task? Figure 15 in the appendix show that the single shared learning rate has a huge impact on entropy and overall performance.
3. The text claims that the policy and Q-value converge to local optimums, and then immediately contradicts this statement by noting that the policy re-rises in the later stage of learning. If the policy is at a local optimum, its entropy could not change. The Q-value curve also suggests that the critic network is still changing at the end of the run. Given a comment in the appendix, these lines are probably also smoothed, is that correct? If so, the networks are actually changing even more than the plots suggest. Is there evidence for this claim of convergence?

In Section 4.2 how are the true values computed for Figure 3 and why is there confidence shading? Are the true values not exact? I think the approximate and true values for each individual run should be compared together so it is unclear how to interpret these plots. Plotting the error rather than the values could alleviate this issue.

I am unconvinced that all of discrete SAC's tuning options were explored before introducing another loss component and hyperparameter. Figure 15 confirms that the shared learning rate has a massive impact on entropy and overall performance. What if the $\alpha$ learning rate is tuned separately? What if $\alpha$ is set to a fixed value, or if that value is annealed over time? There should be a relationship between the learning rate of $\alpha$ and the value of $\beta$ and I would want to see that explored. Since $\alpha$ is the temperature parameter, there is a formal relationship between $\alpha$ and how far the policy update step can move the policy from the previous policy (and therefore how much the entropy can change) in terms of the KL divergence, which we can see in the analysis of mirror descent with a KL divergence regularizer. Does the method proposed by this paper merely trade away having to carefully tune $\alpha$ in return for having to carefully tune $\beta$?

A general concern about the experimental results presented in this paper is that the results are very noisy and variable for only performing three repetitions. In many of the figures, I am uncertain that the observed patterns are actually reliable. Part of the difficulty is that the figures plot confidence as shaded regions (though exactly what these shaded regions is never explained, as far as I can tell). Three runs is not enough to show uncertainty as confidence intervals. Confidence intervals require a variance estimate, which almost always requires more than three runs to be accurate. Please plot each individual run instead, at least in the appendix. For the main plot figures when there are too many competing methods to show all three runs from each, plot the "median run", that is, find the runs for a given method with the largest and smallest final or average returns (or both if they are different and each analysis provides different insights) and exclude them. Then plot only the remaining run.

The Q-clipping that this paper presents is unsafe from a learning and optimization perspective. The clipping not only prevents incentivizing the critic network from making predictions more than $c$ away from those of the target network, it zeros out gradients whenever the critic network makes predictions $c$ away from those of the target network. That means that if $Q \ge Q' + c$ or $Q \le Q' - c$ on all the data points, $Q$ is a local minimum. While the network may not be incentivized to reach such a state, if it does reach such a state, all learning in the critic would stop until the target network is updated and the predictions would likely be bad.

Discrete SAC + penalty + avg-q appears to perform much worse than Discrete SAC + avg-q, but this failure is not addressed. Why does this failure occur?

I do not understand how Figure 7 shows an improvement over discrete SAC. Most of the plots look qualitatively similar to me. The main difference I see is that the plot after ten million steps is more shallow that discrete SAC's but why is that better? All of them look nearly convex and both methods look to me like they have a comparable number of saddle-points and are of similar "complexity".

The Honor of Kings case study is poorly developed and has multiple issues.
1. Saying that Honor of Kings is "the world's most popular MOBA" is an unnecessarily hyperbolic claim.
2. Stating that Honor of Kings is a "popular testbed for RL research" when all three citations for this claim share many of the same authors is disingenuous.
3. This paper proposes that discrete SAC specifically suffers from issues that do not befall SAC in continuous action domains. If that is true, why discretize the direction of movement and skill? Honor of Kings seems like an odd choice to illustrate improvements to discrete SAC when its natural action representation is at least partially continuous.
4. How should we interpret the ELO scores? How were they computed? Who were the opponents? How many matches were played? What does 1343 ELO play look like compared to 430 in this game?

---

> ### Author Response · Authors · 2024-09-04
>
> Thank you for your comments on the manuscript! We provide detailed clarifications for all your concerns below, replace smoothed curves to avoid misunderstanding, incorporate modifications and additional results in the appendix and attachment following your advice (shown in orange in the revised paper).
>
> **Q1: Regarding the failure mode analysis in Section 4.1: How are the rewards "deceptive"? How are the policy and Q-value converge to local optimums? How to interpret the increase of episode length and the number of steps with reward in Figure 2?**
>
> A1: We are sorry for the misunderstanding. For the term "local optimum", we mean that **the agent tends to adopt a short-sighted strategy, failing to discover long-term policies with greater payoff potential**, rather than the network itself stagnating in its updates. Ideally, the agent would consume all objects, recognize that touching lyres results in a loss of life, and adjust its policy to aquire scoring and avoidance skills simutaneously. However, due to the presence of policy entropy in discrete SAC, we observe that the agent struggles to achieve this type of learning.
>
> Recall the optimization objective of policy: $J_{\pi}(\phi)=E_{s_{t} \sim D}[E_{a_{t} \sim \pi_{\phi}}[\alpha \log (\pi_{\phi}(a_{t} \mid s_{t}))-Q_{\theta}(s_{t}, a_{t})]]$  (Equation 8). In early stage, the agent quickly obtain short-term rewards through specific action pattern (e.g. collect all objects without avoidance in Asterix). **The Q value increate too rapidly that it diminishes the effectiveness of policy entropy constraint.** As a result, the agent's policy is rigid, falling into the trap of continuously exploiting short-term rewards - thus entering a local optimum. It is only after the increase in Q values slow down that entropy constraint gradually takes effect, allowing the agent to explore to some extent. **However, both the Q values and scores decline during this phase, indicating that escaping this trap is lengthy and ultimately ineffective.**
>
> We aim to prevent the agent from such situation. In experiments, we observe that by adding entropy penalty, the policy entropy change is smaller and smoother, and there is a significant improvement in both Q values and final scores (in Figure 4).
> We also include a comparison of SDSAC and DSAC in terms of episode length and number of steps with rewards in Appendix A.2 (Figure 9). **After 2e6 steps, the algorithm with entropy penalty demonstrates significant longer episode lengths and more reward steps compared to discrete SAC. This indicates that the entropy penalty helps the agent learn both scoring and avoidance skills, leading to continued performance improvement.**
>
> Additionally, Figure 2 is indeed smoothed, which we have replaced with the original curves. **All plots in the paper are now raw data obtained in the experiment (except Figure 8 which we have annotated in the caption).** We apologize for this mistake.
>
> **Q2: Why are these failure mode specific to discrete SAC?**
>
> A2: The reason SAC does not exhibit these failure modes in continuous environments is twofold. First, SAC employs the reparameterization trick, fitting actions with a Gaussian distribution. The use of tanh function in the reparameterization trick results in a smoother policy distribution with reduced variance,  allowing it to adapt to deceptive rewards without sacrificing policy diversity. Second, in continuous environments, actions that deviate slightly from the best response have minimal impact on the outcome, whereas in discrete settings, different actions can have entirely distinct meanings. Therefore, our analysis primarily focuses on the challenges SAC faces in discrete environments.
>
> To validate this point, we test SAC in the MuJoCo environment (see Figure 16 in Appendix B.1). **The curves indicate that in MuJoCo, the SAC algorithm does not encounter local optimum issues; policy entropy changes are minimal and gradual, while the scores stadily increase. This suggests that SAC does not face the problems described in the paper when applied to continuous tasks.**

---

> ### Author Response · Authors · 2024-09-04
>
> **Q3: A general concern about the experimental results presented in this paper is that the results are very noisy and variable for only performing three repetitions.**
>
> A3: We apologize for the confusion. We select three seeds and plot the graphs following prior works such as PPO[1]. The curves represent the mean value, while the shaded area indicate the 95% confidence intervals, which is a standard practice in related research.
>
> [1] Schulman, John, et al. "Proximal policy optimization algorithms." arXiv preprint arXiv:1707.06347 (2017).
>
> **Q4: In Section 4.2 how are the true values computed for Figure 3 and why is there confidence shading?**
>
> A4: The true value in Figure 2 is represented by the unbiased Monte Carlo return: $V_t=R_{t+1}+\gamma R_{t+2}+\gamma^2 R_{t+3}+\cdots+\gamma^{T-t-1} R_T$. We use Monte Carlo return to measure the estimation bias of the Q function. Therefore the true values have shading.
>
> We agree that comparing approximate and true values individually is crucial. Therefore, we have added the curves for three individual runs in Appendix A.3 (Figure 10). The results show that for each individual seed, discrete SAC consistently suffers from an underestimation problem, while using a single Q leads to an overestimation issue.
>
> **Q5: I am unconvinced that all of discrete SAC's tuning options were explored before introducing another loss component and hyperparameter. ... Does the method proposed by this paper merely trade away having to carefully tune α in return for having to carefully tune β?**
>
> A5: We are sorry that we did not make it clear and have restructured the hyperparameter analysis in Appendix B.2. We first compare the performance of different α learning rate in Figure 19. Although smaller learning rate (1e-6 & 1e-7) allow for some degree of control over policy entropy, the overly conservative updates make it diffucult for the agent to learn effectively, resulting in minimal growth in the final scores. Figure 20(a) compares different values of α, further demonstrating that tuning these parameters does not improve the agent's performance.
>
> In our paper, we also compared our method with algorithms that incorporate alpha annealing, such as TES-SAC. As shown in Tables 1 and 2, our method outperforms these approaches.
>
> We additionally include a comparison in the Asterix environment between different β values and the optimal parameter combination (Figure 20(b)). It is evident that, **across various β values, SDSAC consistently outperforms DSAC.** This indicates that the entropy penalty is a better and more balanced constraint than merely limiting the extent of policy updates.
>
> **Q6: The Q-clipping that this paper presents is unsafe from a learning and optimization perspective.**
>
> A6:  We initially observe an underestimation issue with DSAC and propose the average Q method. However, this introduce the potential for overestimation, prompting us to draw inspiration from PPO[1] and implement Q-clipping. Similar to PPO, clipping data with $Q \geq Q' + c$ or $Q \leq Q' - c$, while reducing sampling efficiency to some extent, significantly enhances training stability and improves final performance. This aligns with our goal of a more stable DSAC algorithm.
>
> [1] Schulman, John, et al. "Proximal policy optimization algorithms." arXiv preprint arXiv:1707.06347 (2017).

---

> ### Author Response · Authors · 2024-09-04
>
> **Q7: Discrete SAC + penalty + avg-q appears to perform much worse than Discrete SAC + avg-q, but this failure is not addressed. Why does this failure occur?**
>
> A7: In our experiment, such scenario only occurs in the Kangaroo game. In the other five environments, combined use of entropy penalty and double averaged clipped Q consistently yields the best performance.
>
> We analyze the reason behind this phenomenon in Kangaroo. It has two distinct characteristics:
>
> (1)**There are no deceptive rewards.** It is extremely difficult to score in the early stages, and the player is easily hit by bullets, resulting in the loss of a life. Therefore, the effectiveness of the entropy penalty is limited.
>
> (2)**The environment discourages long-horizon strategies.** Accoring to the game rules, if the player does not rescue the baby kangaroo within a certain time, a life is automatically lost.
>
> These factors make the entropy penalty less effective in leading the agent to focus on long-term gains, while the double averaged clipped Q is more effective in preventing passive exploration.
>
> We provide the recordings of the agents' gameplay for both algorithms in the attachment. There is a distinct difference in the policies: the agent combining both techniques (Kangaroo_entropy+avgq.gif) not only has strong avoidance abilities but also can score by hitting bullets; the agent that only applies avg-q (Kangaroo_avgq.gif) has weaker avoidance skills, but its more aggressive strategy allows it to score significantly faster by directly hitting monkeys.
>
> A broader investigation shows that **out of the 20 games we tested, 2 games have these characteristics (Frostbite, Kangaroo).** Whereas in other environments, applying the entropy penalty improves the agent's long-term focus, and using double average clipped Q positively impacts the agent's active exploration.
>
> **Q8: I do not understand how Figure 7 shows an improvement over discrete SAC.**
>
> A8: Figure 7 compares the loss surface of our method with that of DSAC. Our method's surface is significantly shallower, with fewer saddle points and no prominent local minima, all of which contribute to a more favorable optimization process during training.

---

> ### Author Response · Authors · 2024-09-04
>
> **Q9: Regarding the Honor of Kings case study.**
>
> **Q9.1: Saying that Honor of Kings is "the world's most popular MOBA" is an unnecessarily hyperbolic claim. Stating that Honor of Kings is a "popular testbed for RL research" when all three citations for this claim share many of the same authors is disingenuous.**
>
> A9.1: Thank you for your constructive suggestion. We have adjusted the corresponding statements to address the overclaiming issue.
>
> **Q9.2: Why discretize the direction of movement and skill?**
>
> A9.2: This is actually an inherent setting of the Honot of Kings testing environment[2]. In fact, other similar MOBA game environments used for reinforcement learning research, such as DoTA[3] and StarCraft[4], also discretize actions and are designed as a discrete action space.
>
> [2] Ye, Deheng, et al. "Mastering complex control in moba games with deep reinforcement learning." Proceedings of the AAAI Conference on Artificial Intelligence. Vol. 34. No. 04. 2020.
>
> [3] Berner, Christopher, et al. "Dota 2 with large scale deep reinforcement learning." arXiv preprint arXiv:1912.06680 (2019).
>
> [4] Vinyals, Oriol, et al. "Grandmaster level in StarCraft II using multi-agent reinforcement learning." nature 575.7782 (2019): 350-354.
>
> **Q9.3: How should we interpret the ELO scores? How were they computed? Who were the opponents? How many matches were played? What does 1343 ELO play look like compared to 430 in this game?**
>
> A9.3: The ELO rating system is a widely-used mechanism for assessing the relative skill levels of players (or agents), commonly applied in chess, competitive games, and other adversarial environments. In our study, the ELO ratings of agents are calculated through the following process:
>
> 1. Each agent is assigned a fixed initial ELO rating $R_{base} = 400$.
>
> 2. Before agent A competes against agent B, the expected score for each agent is calculated based on their current ELO ratings:
> $$E_A=\frac{1}{1+10^{\left(R_B-R_A\right) / R_{base}}}$$
>
> 3. The ELO ratings are updated based on the outcome of the match between A and B, where $S_A$ is the actual score and $K$ is a constant:
> $$R_A^{\prime}=R_A+K \cdot\left(S_A-E_A\right)$$
>
> 4. Through multiple matches among various agents, the ELO ratings are adjusted according to the results of these matches. The final ELO ratings reflect each agent's relative strength compared to the others.
>
> For more detailed ELO calculation formulas, refer to[6]:
>
> [6] Elo, Arpad E. The Rating of Chessplayers, Past and Present. Arco Pub., 1978.
>
> In our experiment, we conducted **48 one-on-one matches for each of the 6 agents (SDSAC-24h, SDSAC-36h, SDSAC-48h, DSAC-24h, DSAC-36h, DSAC-48h), resulting in a total of 720 matches.** The suffix "h" denotes the training time for each agent.
> For instance, in the match between SDSAC-48h (ELO rating 1343) and DSAC-24h (ELO rating 430), the results were as follows: SDSAC-48h achieved **35 wins, 7 draws, and 6 losses, with a win rate of 72.92%.**
>
> We have uploaded a video of one of the matches between SDSAC-48h and DSAC-24h, named "dsac24h_battle_sdsac48h.mov", as an attachment.  In the video, the red agent represents SDSAC, and the blue agent represents DSAC. It can be observed that the red agent's current K/D ratio is 3/1 and that it has successfully bypassed the blue agent's defensive tower range to eliminate the blue agent, indicating that the playing capability of SDSAC-48h significantly surpasses that of DSAC-24h.
>
> Another GIF named "dsac48h_battle_sdsac48h.gif" represents the match between the blue agent SDSAC-48h and the red agent DSAC-48h. From this battle, it can be observed that the blue agent has a higher skill hit rate than red agent.

---

> ### Author Response · Authors · 2024-10-11
>
> Due to the adjustment of the appendix structure during the revision process, we have updated the references in the response regarding the appendix paragraphs and figures. We hope this does not affect your reading.

---

### Review · Reviewer_4k4v · 2024-09-07

**Summary Of Contributions:**

This paper takes a careful look at how we might adapt Soft Actor Critic (SAC) to discrete domains. First, the paper explores the standard methods of discretization, and highlights two failure modes of this standard approach that result in training instability and underestimated Q-values. The paper finds that "deceptive rewards" and the use of a double Q-learning estimator contribute to these failure modes. In light of this, the work then proposes a new variant of discrete SAC that is designed to explicitly overcome these two issues. First, to account for training instability, an entropy penalty is introduced. Second, to account for the underestimation issues, an averaged variant of double Q-learning is used. Together, these two pieces combine to form Stable Discrete SAC (SD-SAC), a new variant of SAC. The remainder of the paper contains extensive experiments that evaluate SD-SAC in a wide variety of domains, including Atari and large-scale MOBA.

**Audience:**

Yes

**Broader Impact Concerns:**

No broader impact concerns.

**Claims And Evidence:**

Yes

**Requested Changes:**

Writing Suggestions:

High-level:
- I suggest removing all colors from the text throughout the paper. I found the blue and oranger text made the paper much harder to read.
- Move Algorithm 1 pseudocode to Section 5.
- Make figures larger.
- Take a careful pass through your equations and notation, with some of the below suggestions in mind.

Low-level, section by section:

Intro:
- "achieve exploit-explore trade-off" --> "balance exploitation and exploration"
- "suggested by (Christodoulou, 2019)" --> "suggested by Christodoulou (2019)"
- "resulting in the discrete version of SAC, denoted as discrete SAC (DSAC)" --> " resulting in discrete SAC (DSAC)"
- "fails also include the absence of policy update constraints" --> "fails also due to the absence of policy update constraints"
- "we propose Stable Discrete SAC" --> "we propose Stable Discrete SAC (SD-SAC)"

 Preliminaries:
- If the action space is discrete, then the Shannon entropy of $\pi(\cdot \mid s)$ would be better expressed as a sum, rather than an integral.
- "bellman" --> "Bellman"
- Equation 1 uses a finite horizon, undiscounted expected (regularized) return. But later, discounts are used. It would be helpful to clearly establish your setting, first.
- Terms are bolded unexpectedly (Equation 9), can you clarify why this is the case in the text?

Failure Modes
- "with the Q-learning iteration" --> "with Q-learning iteration"
- Since your new method builds on the double Q-learning trick, it might be valuable to include slightly more detail about what it is (Section 4.2).
- Can you move Algorithm 1 to Section 5?

Case Study in Honor of Kings
- Since this is only a few paragraphs, I believe you can make this a subsection of Section 6 rather than its own section.

**Strengths And Weaknesses:**

[STRENGTHS]:
- The paper studies what I take to be an important question: SAC has become one of the standard algorithms of the RL community, and this paper gives a careful examination of how it has been adapted to handle discrete actions. The two failure-modes identified are convincing, and they naturally give rise to two modifications that underly SDSAC. In this sense, I believe the approach could be readily adopted by many experimentalists looking to use SAC in discrete domains.
- The experiments are broad and deep. There are significant baselines, domains, and analysis, and the evidence presented is convincing.
- The paper is well organized and does a good job of walking through the narrative that (1) SAC should be adapted to discrete action domains, (2) but it suffers from some issues, so (3) we need some fixes. I am really sympathetic to the story, and find the main contributions to be valuable and I believe will have an audience in the TMLR community.

[WEAKNESSES]
- There are aspects of the writing that could be improved along two axes. First, some aspects of the introduced notation and math are slightly unclear. I include some suggestions in the "Requested changes" section below, but I encourage the authors to take a careful pass through any introduced equations and ensure that all notation has been properly and consistently introduced. Second, parts of the paper are hard to read, such as the brightly colored blue blocks of text. This is an easy fix: I suggest removing all of the blue and orange colors from the text blocks.
- Related to colors: I found the colored series in the plots quite difficult to read. It is in part due to the fact that the figures are so small: can you make them larger? However, the actual colors themselves are extremely similar (at least to my eye), so I had a hard time differentiating between the blue and light purple, for instance.

Overall I believe this paper presents a valuable contribution: the main claims are supported by evidence, and there will be an audience for the work.

---

> ### Author Response · Authors · 2024-09-23
>
> Thank you very much for your positive comment on our paper and interest in its importance! We are also grateful for the useful advice you have provided regarding the details of writing.
>
> Based on your recommendations, **we have revised the language and notation** throughout the paper to ensure greater precision and accuracy. This includes changing the integral in Equation 2 to a sum, incorporating a discount factor into Equation 1, removing the incorrect bold formatting from Equation 9, repositioning the pseudocode, and making several other adjustments to the text.
>
> To ensure coherence, **we have also included a basic introduction to double Q-learning in Section 4.2.** To address the issue of overestimation in DQN, double Q-learning was proposed. This approach mitigates the problem by employing two independent Q-networks, and using the minimum value between them as the final Q-value. The concept was initially introduced by Double DQN in the discrete domain. In the continuous domain, TD3 and SAC also adopt clipped double Q-learning to mitigate overestimation, making it a favored technique across various reinforcement learning algorithms.
>
> Regarding the issue of text color, we used different color in the manuscript to denote the revision history. Specifically, blue was used to indicate changes made since the previous submission, and orange was used to highlight modifications made during the current review phase. **We have removed the blue color; however we retain the orange markings temporarily to assist other reviewers in understanding the latest revisions.** We ensure that all color markings will be removed in the camera-ready version to adhere to publication standards.
>
> Regarding the figures in the paper, we appreciate your suggestions for enhancing their readability. **We have enlarged the figures and standardized the colors of the curves** to make them more accessible and easier to read.

---

### Review · Reviewer_roTL · 2024-10-02

**Summary Of Contributions:**

The paper proposes a new improvement on Discrete Soft Actor-Critic (SAC) algorithms. After a detailed study of the vanilla SAC and outlining two main issues with it - training instability and pessimistic exploration - the paper presents its solution to these problems based on an entropy penalty term and a variant of double Q-learning. It also provides an extensive empirical study of the performance in different game environments.

**Audience:**

Yes

**Claims And Evidence:**

Yes

**Requested Changes:**

See weaknesses section.

Also noticed a few typos:
- "Adaptation" in the abstract
- H(\pi(. | s_t)) in the definition of SAC (p3 section 3)

**Strengths And Weaknesses:**

Strengths:
- Paper is well structured and clearly written
- Nice investigation of the vanilla SAC shortfalls
- New method is interesting and empirical results look promising

Weaknesses:
- Following up on the comments in the previous round of review, I believe the investigation of the 2 issues outlined in section 4 could still benefit from more detailing and testing, e.g. in a different environment that Asterix

---

> ### Author Response · Authors · 2024-10-11
>
> We greatly appreciate your positive feedback on our revision and your recognition of the paper!
>
> Regarding the investigation of the two issues mentioned in Section 4, we agree that diversity in the testing environment is essential. **In the 20 Atari environments selected in our paper, we found that 4 of them have significant issues with deceptive rewards, namely Asterix, Assault, Jamesbond and MsPacman.** This led to the phenomenon of local optima, as discussed in our paper, indicating that such situation is not an isolated case.
>
> Beyond the in-depth analysis and experiments on Asterix presented in the main text, **we have provided a detailed analysis in Appendix A.5 of three additional Atari environments (Assault, Jamesbond, and MsPacman).** This includes an identification of the sources of deceptive rewards and the performance curves of vanilla DSAC during the training process. **To better illustrate the effect of our approach, we have added comparisons of DSAC, SD-SAC, and KL-penalty in these three additional environments in Appendix  A.5.3.** The results show that the entropy penalty introduced by SD-SAC consistently limits the rapid decline in entropy during the early stages of training. This helps prevent the learning process from converging to local optima, ultimately enhancing the agent's performance.
>
> **Additionally, we extended our analysis to continuous domain tasks using the MuJoCo environment, to determine the boundaries of the issue presented in the paper. Our experiments revealed that the SAC algorithm in continuous domains does not encounter the same problems observed in discrete domains.** This may be attributed to the reparameterization trick used in continuous domain SAC algorithms and the inherent characteristics of continuous environments, which allow the algorithm to better adapt to deceptive rewards. The detailed experimental results and analyses for these continuous domain tasks are provided in Appendix B.1. All modifications made in this round of review are highlighted in orange.
>
> We are also grateful for your suggestions regarding the typos in the paper. **We have fixed these two typos in the updated version.**

---

### Decision · Action_Editor_wUQs · 2024-11-07

**Recommendation:** Accept with minor revision

**Comment:**

Overall I think the paper makes a decent contribution to RL research. The two proposed tricks perform well under the present settings and may be used more broadly in future work. The paper improves over the previous TMLR submission and gave more convincing explanations to the observed phenomena than the previous version.

That said, there are many remaining concerns from the reviewer which the authors must address in their final version:
* More careful statistical significance analysis of the results, such as plotting existing runs with individual runs shown (with matching seeds for different algorithms) similar to Figure 10, or (optionally) running more seeds
* Details about the Honor of Kings experiment mentioned in the rebuttal
* (Optionally) more comprehensive tuning of $(\alpha, \beta)$ in the entropy regularization, and justifications of the choices made in the main text experiments

**Audience:**

The paper would be of interest to the RL community.

**Claims And Evidence:**

This paper identifies two issues with the Discrete Soft-Actor Critic algorithm for reinforcement learning: "Unstable Coupling Training", and "Pessimistic Exploration". The paper proposes an entropy regularization trick and a Q clipping trick to address the two issues correspondingly. Experiments show that the tricks address these issues and improve performance on both Atari games and the Honor of Kings game.

---

> ### Author Response · Authors · 2024-11-20
>
> Thank you for your support of our work, your insightful advice and comprehensive efforts during the reviewing process. Based on your suggestions, we have made the following changes to the revised manuscript:
>
> - 1.Regarding the statistical significance analysis of the results, similar to Figure 10, **we plot the individual run curves for two other figures (Figure 11 and 12) during our major analysis (Figure 4 and 5)**. Results show that our SD-SAC, on each seed, demonstrates smoother and more gradual changes in policy entropy as well as more accurate Q-value estimation compared to baselines. This further strengthens the validity of our analysis of the problem and the effectiveness of the proposed approach.
>
> - 2.We have revised the discussion in Section 6.5 to include **additional details of the Honor of Kings environment and experimental results as provided in the rebuttal.** Additionally, we have **added a section in the appendix introducing the ELO rating system** to help readers better understand the implications of the results.
>
> - 3.**We provide a comprehensive hyperparameter analysis in appendix B.2.** By experimenting with different alpha and learning rate, we identify the performance upper bound of discrete SAC, then we run SD-SAC using different beta values, SD-SAC consistently exceeds this upper bound, proving it as a better method. **We have included a brief description of this analysis in section 6.3 of the main text to enhance the completeness of the paper's structure.**
>
> We thank all reviewers for their constructive comments, which have helped us improve the quality of our work. If you have further comments or requested revisions, please let us know!